# Fungal melanin suppresses airway epithelial chemokine secretion through blockade of calcium fluxing

Jennifer L. Reedy [1,2,10], Kirstine Nolling Jensen [1,2,10], Arianne J. Crossen [1], Kyle J. Basham[1], Rebecca A. Ward [1], Christopher M. Reardon[1], Hannah Brown Harding [1,2], Olivia W. Hepworth[1,2], Patricia Simaku [1], Geneva N. Kwaku[1], Kazuya Tone[3,4], Janet A. Willment[3,5], Delyth M. Reid[3], Mark H. T. Stappers[3,5], Gordon D. Brown [3,5], Jayaraj Rajagopal [6,7,8,9] & Jatin M. Vyas [1,2] ✉

Respiratory infections caused by the human fungal pathogen *Aspergillus fumigatus* are a major cause of mortality for immunocompromised patients. Exposure to these pathogens occurs through inhalation, although the role of the respiratory epithelium in disease pathogenesis has not been fully defined. Employing a primary human airway epithelial model, we demonstrate that fungal melanins potently block the post-translational secretion of the chemokines CXCL1 and CXCL8 independent of transcription or the requirement of melanin to be phagocytosed, leading to a significant reduction in neutrophil recruitment to the apical airway both in vitro and in vivo. *Aspergillus*-derived melanin, a major constituent of the fungal cell wall, dampened airway epithelial chemokine secretion in response to fungi, bacteria, and exogenous cytokines. Furthermore, melanin muted pathogen-mediated calcium fluxing and hindered actin filamentation. Taken together, our results reveal a critical role for melanin interaction with airway epithelium in shaping the host response to fungal and bacterial pathogens.

A*spergillus fumigatus* is the most prominent respiratory fungal pathogen and causes a spectrum of clinical manifestations ranging from allergic disease to severe invasive infections. Despite the ubiquity of this fungal pathogen in the environment, only ~10% of immunocompromised patients (*e.g.*, neutropenic or allogenic bone marrow transplant recipients) develop invasive aspergillosis (IA), indicating that other factors play a significant role in determining the true risk for this infection[1]. In addition to infection associated with the immunocompromised state, there is an increased risk of IA following pulmonary viral infections (*e.g.*, influenza, SARS-CoV-2) and in those with underlying lung disease (*e.g.*, asthma, cystic fibrosis [CF], chronic obstructive pulmonary disease [COPD])[2–5]. The mortality from *Aspergillus*-related pulmonary disease remains unacceptably high (>50%), coinciding with elevated rates of *Aspergillus* multidrug resistance[6–8]. Despite this, the mechanistic interactions governing the invasion of *Aspergillus* at its first point of contact with the host, the airway epithelium, remain poorly understood.

[1]Department of Medicine, Division of Infectious Diseases, Massachusetts General Hospital, Boston, MA, USA. [2]Harvard Medical School, Boston, MA, USA. [3]Aberdeen Fungal Group, University of Aberdeen, Institute of Medical Sciences, Foresterhill, Aberdeen, United Kingdom. [4]Division of Respiratory Diseases, Department of Internal Medicine, The Jikei University School of Medicine, Tokyo, Japan. [5]MRC Centre for Medical Mycology, University of Exeter, Exeter, United Kingdom. [6]Center for Regenerative Medicine, Massachusetts General Hospital, Boston, MA, USA. [7]Division of Pulmonary and Critical Care Medicine, Department of Medicine, Massachusetts General Hospital, Boston, MA, USA. [8]Harvard Stem Cell Institute, Cambridge, MA, USA. [9]Klarman Cell Observatory, Broad Institute of Massachusetts Institute of Technology and Harvard, Cambridge, MA, USA. [10]These authors contributed equally: Jennifer L. Reedy, Kirstine Nolling Jensen. ✉e-mail: jvyas@mgh.harvard.edu

In immune cells, activation of pattern recognition receptors (PRRs), such as the C-type lectin receptor (CLRs), Toll-like receptors (TLRs), NOD-like receptors (NLRs), and Rig-I-like receptors, mediate fungal recognition of cell wall components and subsequent host responses[9]. The cell wall of *Aspergillus* is primarily composed of polysaccharides, including galactosaminogalactan, galactomannan, β−1,3 glucan, β−1,4 glucan, and chitin[10]. Most humans are exposed to *Aspergillus* through the inhalation of conidia, the reproductive propagules of the fungus, which range from 2-3 μm in size and are easily aerosolized in the environment[11,12]. In addition to the typical cell wall components, *Aspergillus* conidia are surrounded by a hydrophobic rodlet and 1,8-dihydroxy naphthalene (DHN) melanin layers[10,12]. Once inhaled by the host, conidia swell and shed these outer layers, exposing the carbohydrate matrix of the cell wall. In immune cells, fungal melanin possesses the remarkable capacity to absorb reactive oxygen species (ROS) and blunt pro-inflammatory cascades, serving as a key virulence factor for many pathogens, including *Aspergillus*[13–19]. Amelanotic strains of clinically relevant fungal pathogens lose their capacity to mount successful infections in mammalian host model systems, indicating that melanin plays a key and non-redundant role in virulence. Indeed, the presence of melanized *Aspergillus* leads to blockage of phagosome biogenesis[20] and removal of the melanin in the phagosome is required to reprogram macrophage metabolism, promoting an antifungal response[21].

Clinical data and current literature indicate that several immune system components are critical for the recognition and swift clearance of fungal pathogens. Still, the rules governing the inflammatory responses to fungi in the lungs are poorly understood. Past studies have focused on innate and adaptive immune cells but have often overlooked a key player in respiratory infections: the airway epithelium - the primary point of contact for inhaled conidia[22–25]. The conducting airways are sites of *Aspergillus* colonization and invasion[22–25]. Fortunately, in vitro differentiation of primary airway basal cells recapitulates the true diversity of conducting airway epithelium with all of the relevant common (basal, ciliated, club, goblet) and rare (ionocytes, tuft, neuroendocrine) cell types in pseudostratified layers, enabling mechanistic studies ex vivo[26]. Additionally, these cells form polarized barriers and enable the investigation of basolateral and apical epithelial responses when grown at an air-liquid interface (ALI)[27,28]. While the primary human airway epithelial cell (hAEC) ALI model may not fully recapitulate the in vivo environment (*e.g.*, lacks resident immune cells, endothelial cells, and fibroblasts), this model offers the advantages of a highly pliable system to conduct mechanistic studies without compromising relevant human physiology[28–30] and allow us to pinpoint mechanisms specific to the interaction of the epithelium and inhaled conidia where the microbe first encounters host defense. In addition to utilizing this system, we also use the airway epithelial cell line H292, a mucoepitheliod non-small cell lung carcinoma cell line, which has previously been used extensively in modeling airway epithelial responses to bacterial pathogens. While this cell line does not capture the cellular diversity of the airway epithelium, it is well-suited to mechanistic studies due to cellular uniformity. We previously demonstrated a role for melanin in the immune response to *Aspergillus fumigatus* orchestrated by airway epithelium using both primary hAEC and H292 epithelial models[28]. We observed that airway epithelium infected with *A. fumigatus* lacking DHN melanin promoted greater transepithelial migration of neutrophils than wild-type conidia. The mechanism by which melanin modulates epithelial-mediated neutrophil transmigration remains unknown.

In the present study, we leveraged primary hAEC to demonstrate that purified fungal melanin at the apical surface of epithelial cells blocked the migration of neutrophils and blunted the apical secretion of chemoattractant chemokines. We determined that DHN melanin suppression of neutrophil transmigration and chemokines occurred in response to both fungal and bacterial pathogens. Furthermore, L-DOPA melanin from *Cryptococcus neoformans* and other commercially available synthetic melanins inhibited CXCL8 (also known as IL-8) secretion, showing these effects were not unique to DHN melanin, but rather is a preserved property of fungal melanins. We determined that fungal melanin abolished the apical-basolateral CXCL8 gradient as a mechanism to block neutrophil recruitment. We demonstrated that melanin suppressed secretion, but not translation or transcription, of CXCL8 in airway epithelium in a phagocytosis-independent manner. Lastly, we unveiled that melanin exerts effects through dampened calcium fluxing and disrupted actin filamentation in airway epithelium. Together, these data indicated that fungal melanin inhibits epithelial-mediated inflammation by blocking calcium signaling and the secretion of critical pro-inflammatory chemokines.

## Results

### *Aspergillus* melanin inhibits transepithelial migration of primary neutrophils

We previously demonstrated that hAEC induces neutrophil transmigration upon stimulation with *A. fumigatus* wildtype conidia[28]. However, induction of neutrophil transmigration required at least 4 h of infection with *A. fumigatus* in contrast to the bacterial pathogen, *Pseudomonas aeruginosa*, which induced robust neutrophil transmigration within 1 h[28]. Furthermore, *Aspergillus* conidia that lacked the ability to produce DHN melanin (*ΔpksP*) induced neutrophil transmigration within 1h of infection similar to *P. aeruginosa*, suggesting that melanin was blocking hAEC responses to resting conidia[28]. Although prior studies identified MelLec as a receptor for DHN melanin in endothelial cells and myeloid cells[31–33], primary hAECs and H292 cell line lacked this receptor (Supplementary Fig. 1), suggesting that the effects of fungal melanin are exerted through alternative mechanisms. Thus, we hypothesized that melanin could act via two mechanisms to block neutrophil migration: melanin could actively downregulate pro-inflammatory responses or serve as a passive barrier to prevent host access to immunostimulatory epitopes on the fungal cell wall. To test these hypotheses, we performed mixing studies with *A. fumigatus* melanin ghosts and amelanotic *ΔpksP* conidia. Generation of melanin ghosts is a well-described technique in which melanized conidia are biochemically and enzymatically degraded to strip all cellular components except a rigid shell of purified melanin resembling the original size and shape of the conidia[20,34–38]. To ascertain whether melanin coating of conidia was required for its inhibitory function, we stimulated primary hAEC with *ΔpksP* conidia alone or in combination with melanin ghosts at the apical side at a ratio of 5:1 melanin ghosts to *ΔpksP* conidia and then measured neutrophil recruitment across the epithelium (basolateral→apical) (Fig. 1a). Stimulation of hAEC with *ΔpksP* conidia mixed with melanin ghosts dampened neutrophil transmigration when compared to *ΔpksP* alone (Fig. 1b). As expected, melanin ghosts alone did not stimulate neutrophil transmigration.

We next examined if this response seen in primary hAEC extended to the human lung epithelial cell line, H292, which is routinely used to dissect airway responsiveness to bacterial pathogens[39–41]. While these cells do not recapitulate the cellular diversity of the primary airway epithelium, they are well suited to mechanistic studies given their homogeneity, thus we wished to determine if our phenotype could translate to this model system. We observed that infection with *ΔpksP* conidia alone induced more neutrophil recruitment across H292 monolayers compared with *ΔpksP* conidia in the presence of melanin ghosts (Fig. 1c) recapitulating the primary hAECs findings. Since the melanin in these experiments was applied as discrete particles unable to coat the surface of the *ΔpksP* conidia, these results indicate a model where melanin actively downregulates inflammatory responses rather than acting as a passive barrier on the surface of the conidia.

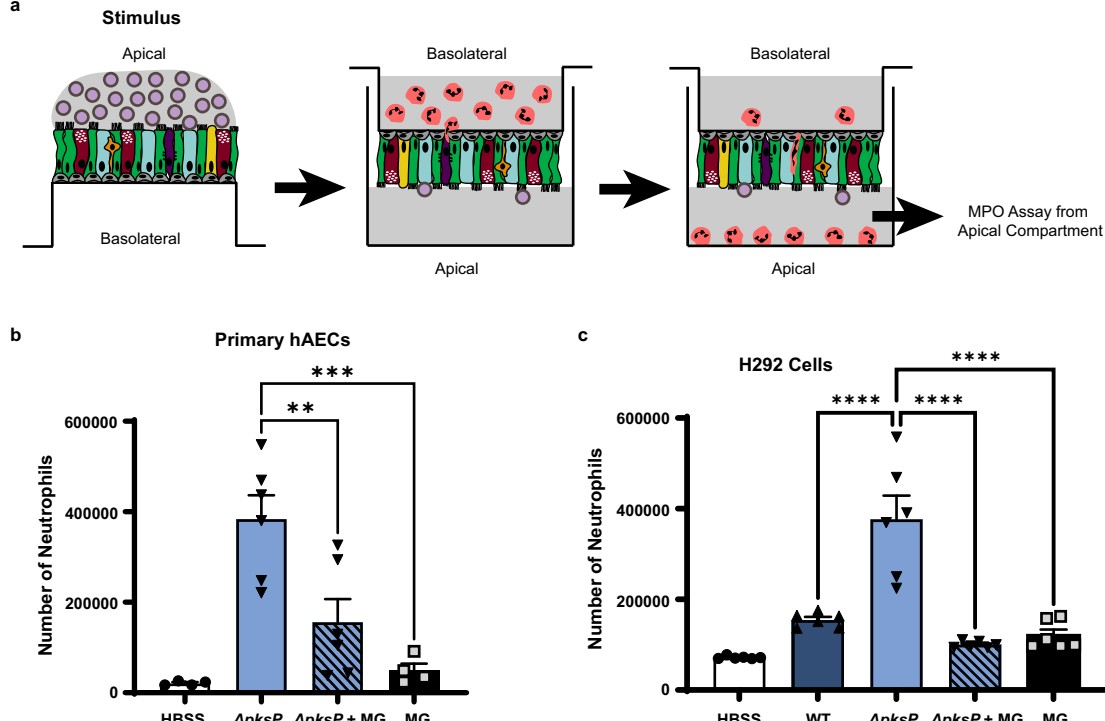

**Fig. 1 | *Aspergillus* melanin inhibits transepithelial migration of neutrophils.**
**a** Schematic of neutrophil migration assay. Neutrophil migration across primary human airway epithelial cells (hAECs) (**b**) or H292 cells (**c**) measured by apical MPO assay following infection (2 h hAECs; 1 h H292) with $1 \times 10^7/cm^2$ wildtype B5233 or *ΔpksP A. fumigatus* conidia in the presence or absence of purified *A. fumigatus* melanin ghosts (MG; $5 \times 10^7/cm^2$). Negative control of media (HBSS) or MG alone was included. Data represents the number of neutrophils as determined by the standard curve. Data are represented as mean ± SEM. N = 4 (primary hAECs controls [HBSS, MG]); N = 6 (primary hAECS infections and all H292 cells) from three experiments; One-Way ANOVA with Tukey's multiple comparisons test; **$p = 0.006$, ***$p = 0.0005$, ****$p < 0.0001$. Source data are provided as a Source Data file.

## *Aspergillus* melanin inhibits secretion of airway epithelial-derived CXCL8 and CXCL1

To understand the mechanism of how neutrophil transmigration was silenced, we performed a Luminex assay using primary human airway epithelium derived from three healthy human donors. Primary hAEC was differentiated at ALI for 16 days prior to stimulation with either media alone (HBSS), *P. aeruginosa* (positive control), three wildtype strains of *A. fumigatus* (CEA10, Af293, B5233), or the amelanotic *A. fumigatus ΔpksP* strain (derived from B5233). CXCL8 and CXCL1 (also known as GROα) are two neutrophil chemoattractants produced by the airway epithelium and were the most abundant chemokines specifically induced on the apical side by the mutant *ΔpksP A. fumigatus* strain, but not by wildtype strains (Fig. 2a, b and Supplementary Figure 2). We confirmed these differences in cytokine expression in H292 cells using both the *ΔpksP* and *ΔpksP* complemented strains (Fig. 2c–d) to validate the use of these cells for mechanistic studies.

To determine if melanin acted through a blockade of epithelial chemokine secretion, we performed mixing studies in which we stimulated with a fixed concentration of *ΔpksP* conidia combined with graded amounts of melanin ghosts. The mixing assay revealed that the addition of melanin ghosts suppressed both CXCL8 and CXCL1 chemokine apical secretion in a dose-dependent manner (Fig. 2e–f). Melanin ghosts alone applied at the highest concentration did not induce chemokine secretion. To confirm that these results were not due to absorption by melanin ghosts of secreted CXCL8, we examined neutrophil migration to the apical compartment in epithelium exposed to melanin ghosts in the presence and absence of exogenous CXCL8. Indeed, despite the presence of melanin ghosts, exposure to CXCL8 triggered robust neutrophil migration (Fig. 2g). These data suggest that melanin ghosts block the secretion of CXCL8 and CXCL1 rather than adsorb extracellular chemokines.

## The inhibitory effects of *Aspergillus* melanin extend to other pro-inflammatory stimuli

Thus far, our experiments suggested that melanin was not acting as a passive protective shield, but directly downregulated pro-inflammatory responses. We next sought to determine if fungal melanin anti-inflammatory actions were specific to fungal pathogens or if they represented a more general host-microbe interaction. *P. aeruginosa* is an important bacterial respiratory pathogen frequently found in co-infections with *A. fumigatus* in patients with CF. Therefore, we examined whether *Aspergillus* melanin ghosts could dampen *P. aeruginosa*-induced airway epithelial inflammation. We infected H292 monolayers with *P. aeruginosa* alone or in combination with melanin ghosts for 6 h then quantified chemokine secretion. The addition of melanin ghosts suppressed CXCL8 (Fig. 3a) and CXCL1 (Supplementary Figure 3a) release in response to *P. aeruginosa* in a dose-dependent manner. To ensure that the melanin ghosts were not interfering with the ability of *P. aeruginosa* to bind to the epithelium, we confirmed that *P. aeruginosa* binding or internalization by the epithelial cells was not altered in the presence of melanin ghosts (Supplementary Figure 3b), suggesting that the difference in chemokine secretion was not due to difference in the ability of *P. aeruginosa* to access the cells. Additionally, we performed an LDH cell viability assay that demonstrated no difference in cell viability in the presence of *Pseudomonas* with or without melanin (Supplementary Figure 3c).

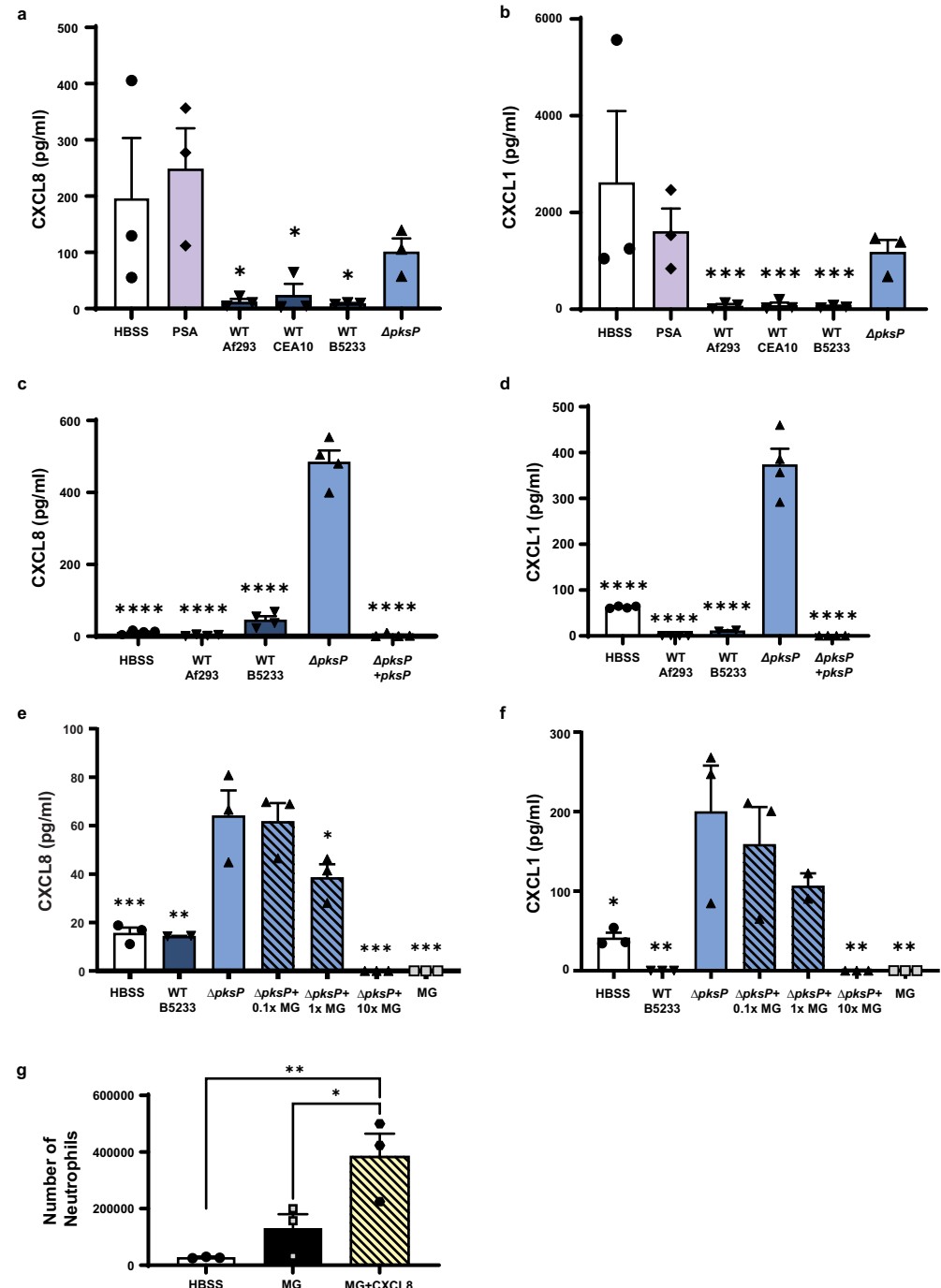

**Fig. 2 | *Aspergillus* melanin inhibits secretion of airway epithelial-derived CXCL8 and CXCL1.** CXCL8 and CXCL1 in apical cell supernatants were measured by Luminex (**a–b**) and ELISA (**c–f**). Data are represented as mean ± SEM. One-way ANOVA with Tukey's multiple comparisons test. (**a–b**) Primary hAECs following 4 h stimulation by media alone (HBSS), *P. aeruginosa* PAO1 strain (PSA), three different wildtype (WT) *A. fumigatus* strains ($1 \times 10^7/cm^2$) or *ΔpksP A. fumigatus* conidia ($1 \times 10^7/cm^2$). N = 3 from three replicates; (a) *ΔpksP* vs Af293 ($p = 0.0168$), CEA10 ($p = 0.034$), or B5233 ($p = 0.0138$); (b) *ΔpksP* vs Af293 ($p = 0.0014$), CEA10 ($p = 0.0150$), or B5233 ($p = 0.0012$). See also Figure S2 (**c-d**) H292 cells were stimulated for 4 h with HBSS, two WT *A. fumigatus* strains, *ΔpksP* conidia, or *ΔpksP* complemented strain ($1 \times 10^7/cm^2$). N = 4 from two replicates; ****p < 0.0001

compared to *ΔpksP*. (**e–f**) H292 were stimulated for 4 h with HBSS, WT B5233 conidia ($1 \times 10^7/cm^2$), *ΔpksP* conidia ($1 \times 10^7/cm^2$), *ΔpksP* conidia mixed with increasing concentrations of *Aspergillus* melanin ghosts (MG; $0.1x = 1 \times 10^6/cm^2$; $1x = 1 \times 10^7/cm^2$; $10x = 1 \times 10^8/cm^2$), or MG alone ($1 \times 10^8/cm^2$). N = 3 from three replicates; (**e**) *ΔpksP* vs HBSS ($p = 0.0002$), B5233 ($p = 0.0004$), *ΔpksP* + 1xMG ($p = 0.0297$), *ΔpksP* + 10xMG ($p < 0.0001$), or MG ($p < 0.0001$); (**f**) *ΔpksP* vs HBSS ($p = 0.0106$), B5233 ($p = 0.0018$), *ΔpksP* + 10xMG ($p = 0.0018$), or MG ($p = 0.0018$). (**g**) Neutrophil migration to the apical compartment across H292 epithelium pre-treated for 1 h with MG ($1 \times 10^8/cm^2$) in the absence and presence of exogenous CXCL8 (3000 pg/mL). N = 3; MG + CXCL8 vs HBSS ($p = 0.0095$) or MG ($p = 0.0416$). Source data are provided as a Source Data file.

TNFα is a known inducer of secretion of CXCL8 by airway epithelial cells, therefore we hypothesized that melanin ghosts also block cytokine-induced CXCL8 in airway epithelial cells. We stimulated H292 epithelium with recombinant human TNFα for 6 h in the presence of absence of melanin ghosts then quantified CXCL8 secretion in supernatants. The combination of melanin ghosts and TNFα stimulation of epithelial cells failed to secrete CXCL8 compared to stimulation with TNFα alone (Fig. 3b).

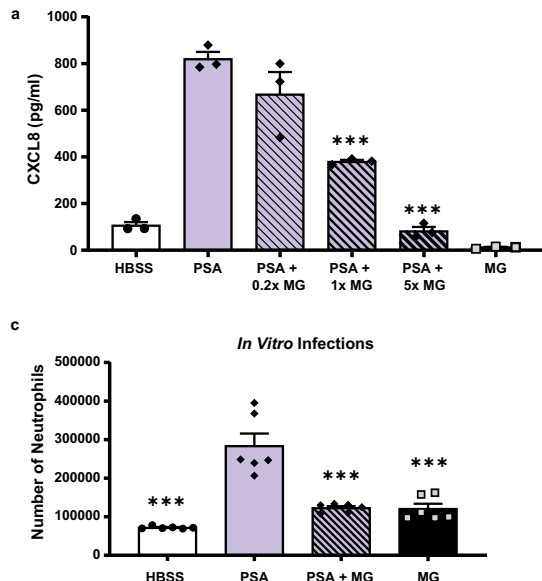
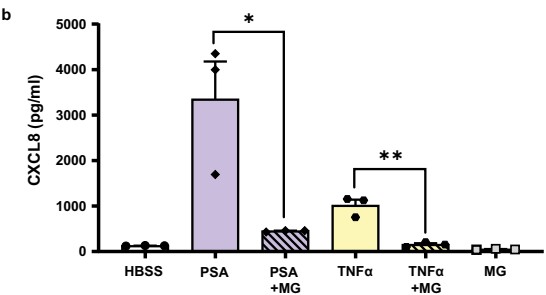
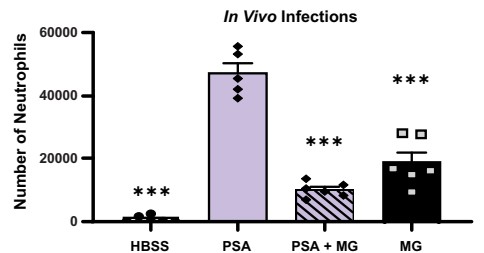

**Fig. 3 | The inhibitory effects of *Aspergillus* melanin extended to other pro-inflammatory stimuli. a** H292 epithelium infected with media alone (HBSS; negative control), *P. aeruginosa* PAO1 strain (PSA) in the absence and presence of increasing *A. fumigatus* melanin ghosts (MG; 0.2x = $2 \times 10^6$/cm²; 1x = $1 \times 10^7$/cm²; 5x = $5 \times 10^7$/cm²) or MG alone for 6 h. Apical CXCL8 secretion in the supernatant was measured by ELISA. *n* = 3 from three replicates; ***p < 0.0001. See also Figure S3. **b** Apical CXCL8 secretion was measured by ELISA following a 6 h stimulation of H292 cells with HBSS, PSA ± MG, TNFα (100 ng/mL) ± MG, or *Aspergillus* MG alone ($5 \times 10^7$/cm²). *n* = 3 from three replicates; *p = 0.0007, **p = 0.0005. **c** Neutrophil migration across H292 epithelium in response to PSA alone or in the presence of

MG or MG alone ($5 \times 10^7$/cm²) was measured by apical MPO assay. Data represents the number of neutrophils as determined by the standard curve. *n* = 6 from three replicates; ***p < 0.0001 compared to PSA. **d** 6–12-week-old C57BL/6 mice were infected via oropharyngeal aspiration with PSA ($5.85 \times 10^5$) with or without *Aspergillus* MG ($5.85 \times 10^6$) for 4 h. The number of neutrophils in the bronchoalveolar lavage fluid was measured by flow cytometry. *n* = 5 for HBSS and *n* = 6 for PSA, PSA + MG, and MG alone for two experiments; ***p < 0.0001 compared to PSA. Data are represented as mean ± SEM. One-way ANOVA with Tukey's multiple comparisons test. Source data are provided as a Source Data file.

## Melanin blocks *Pseudomonas*-induced transepithelial migration of neutrophils

Since co-stimulation of airway epithelium with *P. aeruginosa* and melanin ghosts resulted in a potent decrease of CXCL8 secretion, we examined the role of melanin ghosts in airway epithelial-induced neutrophils transmigration in response to *P. aeruginosa*. While CXCL8 is a known neutrophil chemoattractant that plays a role in epithelial-induced neutrophil transmigration[42,43], recruitment of neutrophils across the epithelium has been linked to other secreted compounds, such as hepoxilin A3, in response to *P. aeruginosa*[44–47]. We stimulated both primary hAEC and H292 with *P. aeruginosa* alone or in combination with melanin ghosts and quantified neutrophil transmigration using a myeloperoxidase assay. The addition of melanin ghosts decreased *P. aeruginosa*-induced neutrophil transmigration (Fig. 3c), demonstrating that melanin blocked all mediators required for neutrophil recruitment to *Pseudomonas*.

To determine if blunting of neutrophil influx occurs in an in vivo model of infection, we infected mice with *P. aeruginosa* alone or in combination with melanin ghosts. As predicted, *P. aeruginosa* induced potent neutrophil recruitment into the airways of mice; however the number of neutrophils in the bronchoalveolar lavage fluid was significantly decreased in the presence of melanin (Fig. 3d). Together, these data indicated that modulation of pro-inflammatory processes by melanin ghosts extends to recruitment of neutrophils to the site of infection in vivo.

## Structurally distinct melanins from diverse fungal sources block airway epithelial CXCL8 expression

Melanins are darkly pigmented molecules created through the polymerization of phenol or indole precursors forming polymers of high molecular weight[48,49]. Despite the ubiquity of melanin pigments across kingdoms (*e.g.*, animals, fungi, insects, helminths), the structure of melanin remains poorly defined. *Aspergillus* and dematiaceous molds

produce DHN melanin catalyzed by the polyketide synthase pathway[14–16,50]. However, some fungal pathogens contain melanin derived from L-tyrosine or L-dihydroxyphenylalanine (L-DOPA) produced through the action of tyrosinase or laccase, respectively[34,35,51–56]. One example of laccase-derived melanin occurs in *C. neoformans*, an environmental yeast that causes human infections including pneumonia and meningitis. L-DOPA-based melanins are structurally distinct from *Aspergillus* DHN melanin. This structural difference impacts host responses as demonstrated by the recognition of DHN, but not L-DOPA, melanin by the CLR, MelLec or Clec1A, identified on endothelial and immune cells[31–33]. To determine if the ability to inhibit the appearance of epithelial-derived CXCL8 in the supernatant was specific to *Aspergillus* DHN melanin, we utilized both *C. neoformans* melanin ghosts and a synthetic tyrosine-based melanin. As strong inducers of epithelial CXCL8, *Pseudomonas* and amelanotic *ΔpksP A. fumigatus* were used to stimulate H292 epithelium in the presence or absence of *C. neoformans* melanin ghosts. Similar to *Aspergillus* melanin ghosts, *Cryptococcus* melanin ghosts suppressed CXCL8 and CXCL1 in the supernatant (Fig. 4a, Supplementary Fig. 4).

In addition to our *C. neoformans* melanin ghosts, we also tested epithelial responses to a synthetic source of tyrosine-derived melanin. We previously created fungal-like particles (FLP) using purified fungal carbohydrates associated with amine-coated polystyrene 3 µM microspheres using either chemical conjugation or adsorption[57,58]. Using this technique, we coated beads to create FLP with synthetic L-DOPA and β-1,3 glucan. β-1,3-glucan is a pro-inflammatory carbohydrate found in most fungal cell walls and was added as an additional control to demonstrate that any effect was due to the melanin and not a result of any coating applied to the microspheres. We stimulated H292 monolayers with *P. aeruginosa* either alone or in combination with unmodified FLP (naked beads), L-DOPA melanin-coated FLP, or β-1,3-glucan-coated FLP. We found that combined treatment with *P. aeruginosa* and L-DOPA-coated FLP led to a reduction in CXCL8

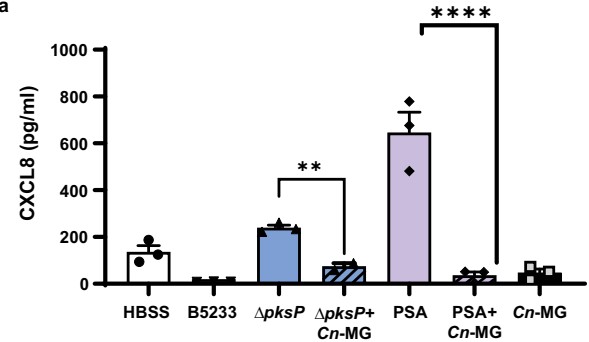

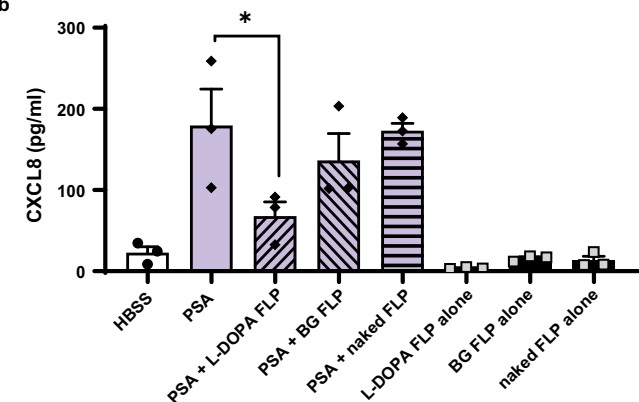

**Fig. 4 | Structurally distinct melanins from diverse fungal sources block airway epithelial CXCL8. a** H292 cells were stimulated for 6 h with wildtype *A. fumigatus* (B5233; $1 \times 10^7/cm^2$), *ΔpksP* conidia ($1 \times 10^7/cm^2$) with or without *C. neoformans* H99 melanin ghosts (*Cn*-MG) ($5 \times 10^7/cm^2$), *P. aeruginosa* PAO1 strain (PSA) with or without *Cn*-MG, or *Cn*-MG alone. CXCL8 apical secretion was measured by ELISA. See also Figure S4. **b** H292 epithelium was infected with PSA in the absence or presence of synthetic L-DOPA-coated FLP ($5 \times 10^7/cm^2$), β–1,3-glucan-coated FLP (BG; $5 \times 10^7/cm^2$), or naked FLP ($5 \times 10^7/cm^2$) for 6 h, then apical CXCL8 secretion was measured by ELISA. Media or coated beads alone were included as negative controls. Data are represented as mean ± SEM. *n* = 3 from three replicates; One-Way ANOVA with Dunnett's multiple comparisons test; *$p < 0.049$, **$p = 0.0004$, ****$p < 0.0001$ compared to stimulation alone (*i.e.*, *ΔpksP* or PSA alone). Source data are provided as a Source Data file.

secretion, while no difference in CXCL8 secretion was observed when *P. aeruginosa* was combined with either unmodified FLP or β-1,3-glucan-coated FLP (Fig. 4b). Taken together, these results demonstrate that the immunomodulatory effect of melanin is not specific to *Aspergillus* melanin, but also occurs in response to *C. neoformans* L-DOPA based melanin and a synthetic tyrosine-based melanin.

## Melanin abolishes the transepithelial gradient of CXCL8 required for neutrophil transmigration

Since epithelial cells are polarized, they secrete effector molecules selectively to either the apical or basolateral compartments or bi-directionally. Neutrophil transmigration to the apical surface (*i.e.*, airway), requires the creation of a chemokine gradient established by differential secretion at the apical and basolateral surfaces[59–61]. To understand whether apical stimulation with melanin affected only apical chemokine secretion or both the apical and basolateral compartments, we stimulated H292 monolayers grown on Transwells with *P. aeruginosa* in the presence or absence of melanin ghosts and collected supernatants from both the apical and basolateral compartments. H292 was chosen to model this interaction because basolateral chemokine secretion by hAECs, even in the presence of *P. aeruginosa* was below the limit of detection. Stimulation with *P. aeruginosa* induced a higher release of CXCL8 on the apical surface of the

epithelium when compared to the basolateral surface (Fig. 5a). The combination of *P. aeruginosa* with *Aspergillus* melanin ghosts significantly decreased both apical and basolateral epithelium CXCL8 secretion when compared with *Pseudomonas* alone. Furthermore, this co-stimulation effectively abolished the apical-basolateral chemokine gradient generated by *P. aeruginosa*.

## Melanin induces durable impact on CXCL8 secretion

Thus far, our experiments examined the impact of melanin ghosts for short durations (6 h or less). Thus, we sought to understand whether melanin stimulation of epithelium produced a lasting effect at the epithelium or was a transient suppression of chemokine release. Melanin ghosts were durable and were not degraded in our cell culture systems as shown in previous studies utilizing melanin ghosts[19,62,63]. We stimulated H292 epithelium with *P. aeruginosa* or TNFα in combination with melanin ghosts for 24 h. After 24 h stimulation, there was a persistent reduction of apical CXCL8 secretion by epithelial cells co-stimulated melanin ghosts compared with either *P. aeruginosa* or TNFα alone (Fig. 5b), supporting the hypothesis that melanin silenced the airway epithelium to prevent inflammation, not simply delaying the initiation of response.

## Melanin suppression of chemokine secretion is independent of phagocytosis of melanin or de novo transcription

To determine the mechanism by which melanin blocked CXCL8 secretion by airway epithelial cells, we first examined whether phagocytosis of melanin ghosts was required. H292 epithelium was pre-treated with cytochalasin D, an inhibitor of actin polymerization that effectively blocks phagocytosis, then stimulated with either TNFα alone or in combination with *Aspergillus* melanin ghosts. Melanin ghosts inhibited TNFα−induced CXCL8 secretion both in the presence and absence of cytochalasin D, suggesting that phagocytosis of melanin ghosts was not required for its anti-inflammatory action (Fig. 6a). Cytochalasin D alone increased basal levels of secretion compared to media alone as shown previously in H292 cells[64].

We next investigated if melanin-induced transcription was required for CXCL8 suppression using the transcription inhibitor, actinomycin D. Following pre-treatments with vehicle control or actinomycin D, epithelial cells were incubated with TNFα, TNFα plus *Aspergillus* melanin ghosts, or melanin ghosts alone. TNFα induction of CXCL8 required CXCL8 transcription[65], therefore as expected addition of actinomycin D blocked TNFα−induced CXCL8 secretion (Fig. 6b). Airway epithelial cells produce basal levels of CXCL8 as demonstrated by the media only controls where there was low, but detectable concentrations of CXCL8. Melanin ghosts suppressed basal levels of CXCL8 secretion by H292 cells. Actinomycin D-mediated inhibition of transcription did not alter melanin ghosts-induced suppression of basal CXCL8 (Fig. 6b). Together, these data revealed that the suppression of CXCL8 secretion was independent of phagocytosis and de novo gene transcription.

## Melanin exerts a post-translational blockade on CXCL8 secretion

To understand further how melanin acted at a cellular level, we sought to illuminate the mechanism by which melanin interfered with CXCL8 secretion by epithelial cells. Therefore, we performed paired experiments, in which we stimulated both primary hAEC cultures (Supplementary Figures 5a and b) or H292 monolayers (Fig. 6c, d) with media alone, wildtype *A. fumigatus* conidia, *ΔpksP* conidia alone or in combination with melanin ghosts, *P. aeruginosa* alone or in combination with melanin ghosts, and melanin ghosts alone. *CXCL8* transcription was induced by all stimuli compared with media treatment alone and no significant difference in transcript levels between our pro-inflammatory stimuli (*ΔpksP* or *P. aeruginosa*) either alone or in combination with melanin, demonstrating that melanin ghosts did not

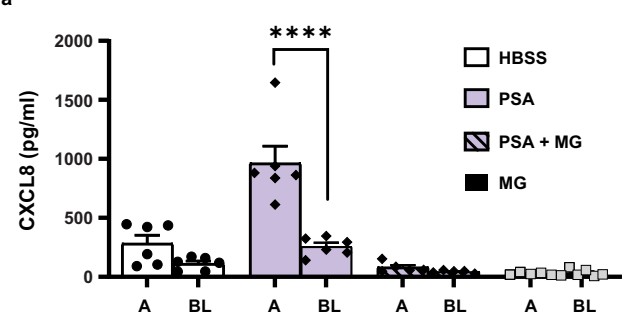

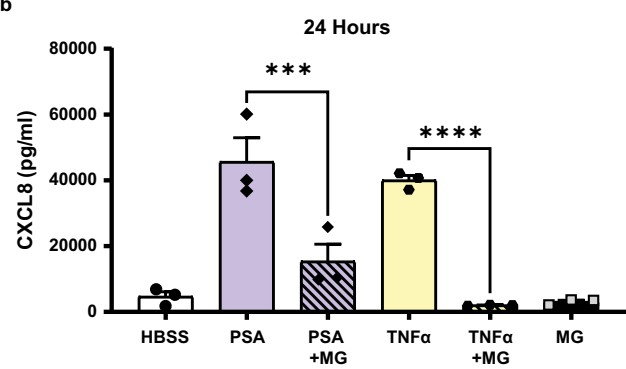

**Fig. 5 | Melanin abolished the transepithelial gradient of CXCL8 required for neutrophil transmigration, with a durable impact on CXCL8 secretion. a** CXCL8 expression in the media from the apical (A) and basolateral (BL) sides of H292 cells was measured by ELISA. H292 cells were infected for 6 h with *P. aeruginosa* PAO1 strain (PSA) and *Aspergillus* melanin ghosts (MG; $5 \times 10^7/cm^2$) together or alone. Media alone (HBSS) was used as a negative control. $n = 6$ (HBSS, PSA, MG BL) or $n = 5$ (PSA + MG, MG A [due to technical error]) from three replicates; two-way ANOVA with Sidak's multiple comparisons test. ****$p < 0.0001$. (**b**) An ELISA assessed apical CXCL8 levels following a 24 h stimulation with PSA or TNFα (100 ng/mL) in the absence or presence of MG ($5 \times 10^7/cm^2$). $n = 3$ from three replicates; one-way ANOVA with Tukey's multiple comparison test. Data are represented as mean ± SEM. ***$p = 0.0011$, ****$p = 0.0001$. Source data are provided as a Source Data file.

block transcription of *CXCL8* (Fig. 6c, and Supplementary Figure 5a). Immunoblotting of epithelial cell lysates revealed increased intracellular CXCL8 in cells treated with melanin ghosts in combination with a pro-inflammatory stimulant (*i.e., ΔpksP* conidia or *P. aeruginosa*) compared to cells treated with the pro-inflammatory stimulant alone (Fig. 6d, and Supplementary Figure 5b). Interestingly, we observed an increase of intracellular CXCL8 protein within cells treated with melanin ghosts, which is in direct contrast to the decrease in extracellular release of CXCL8 demonstrated by ELISA. These data demonstrated that the difference in extracellular CXCL8 protein in epithelial supernatants was not due to a difference in either transcription or translation of the CXCL8 protein. Taken together our results revealed that melanin interfered with CXCL8 signaling by airway epithelium through a post-translational blockade of chemokine secretion.

**Fungal melanin blocked calcium fluxing in epithelial cells**
Given our data suggesting melanin exerts its effects through a transcription/translation-independent mechanism, we hypothesized that fungal melanin mediated chemokine reductions through perturbation of an early signaling pathway- calcium fluxing. To examine if fungal melanin modulated host responses through calcium signaling, we measured calcium fluxing over time in airway epithelium stimulated by ionomycin and calcium chloride or media alone in the presence and absence of melanin ghosts by plate assay (Fig. 7a) and confocal

microscopy (Fig. 7b). As expected, treatment with ionomycin and calcium chloride-induced robust calcium fluxing, with a peak around 10 minutes. Melanin potently muted calcium fluxing to similar levels as unstimulated cells (media alone). Together, these data suggest that fungal melanin modulates epithelial host responses through calcium signaling.

**Melanin disrupts actin structure in airway epithelial cells**
Our results demonstrate chemokine secretion, but not production, is blocked by fungal melanin. To examine whether actin polymerization, which can be mediated by calcium, changes in airway epithelium in the presence of *A. fumigatus* melanin ghosts, we imaged phalloidin-stained epithelial cells pre-treated with *P. aeruginosa* or media alone with or without melanin ghosts. Phalloidin enables visualization of actin filaments (*i.e.*, F actin). In the presence of fungal melanin, epithelial cells showed dampened levels of phalloidin staining (Fig. 8a, b). Using the same experimental groups, we next measured the amount of F and G actin by immunoblotting. We added latrunculin A (an inhibitor of actin polymerization), phalloidin (positive F actin control), and DMSO vehicle as additional controls. These results confirmed reduced F actin levels (Fig. 8c, d), indicating that the fungal melanin disrupts actin polymerization resulting in diminished chemokine secretion in response to inflammatory stimulus.

## Discussion

Humans are exposed to most fungal pathogens, including *A. fumigatus*, through inhalation of fungal particles including conidia, yeasts, or hyphal fragments. Upon inhalation, *Aspergillus* conidia are deposited throughout the host airways, however we lack a detailed understanding of the role that respiratory epithelium plays in directing the outcome of host-pathogen interaction towards clearance, invasion, or allergic disease. Here, we utilized primary, well-differentiated hAEC at ALI and the monomorphic H292 cell line to demonstrate that fungal melanin actively downregulates airway epithelial-driven chemokine responses. Our results demonstrate that *Aspergillus* melanin potently blocks epithelial-driven chemokines via a post-translational inhibition of secretion by muting calcium fluxing and disruption of actin polymerization, resulting in the ablation of the chemokine gradient that silences neutrophil transmigration.

The primary hAEC model is powerful model as it recapitulates the pseudostratified structure and cellular diversity of conducting airway epithelium, enabling study of rare populations and the interplay of epithelial cells in disease[26]. Most epithelial cell types cannot be grown in pure cultures, limiting our understanding of the role of specific epithelial cell subtypes in the pathogenesis of disease. While this system has clear advantages over oncogenic and immortalized cell lines, we also confirmed our phenotype in the the cell line, H292, a mucoepitheliod non-small cell lung carcinoma cell line, as this reagent has previously been used extensively in modeling airway epithelial responses to bacterial pathogens[27,39,40,66–68]. While this cell line does not capture the cellular diversity of the airway epithelium, it is well-suited to mechanistic studies due to uniformity of cellular responses and ability to grow large-scale cultures without a dependence on patient sample quality. Importantly, our results demonstrate that melanin affects both primary hAEC and H292 epithelial similarly, indicating that H292 cells can serve as a relevant model system for mechanistic dissection of melanin sensing and response pathways.

Prior studies identified an impact on the innate immune sensing of fungal pathogens by fungal melanin. Melanin reprogramed the metabolome of macrophages and prevented LC3-associated phagocytosis of *Aspergillus* conidia through sequestration of calcium preventing calcineurin activation[21,37,69,70]. Additionally, melanin is a known inhibitor of ROS production by phagocytes[12–17], which is an important mechanism of pathogen killing by macrophages and neutrophils. Since the airways are the first point of contact in invasive aspergillosis, we

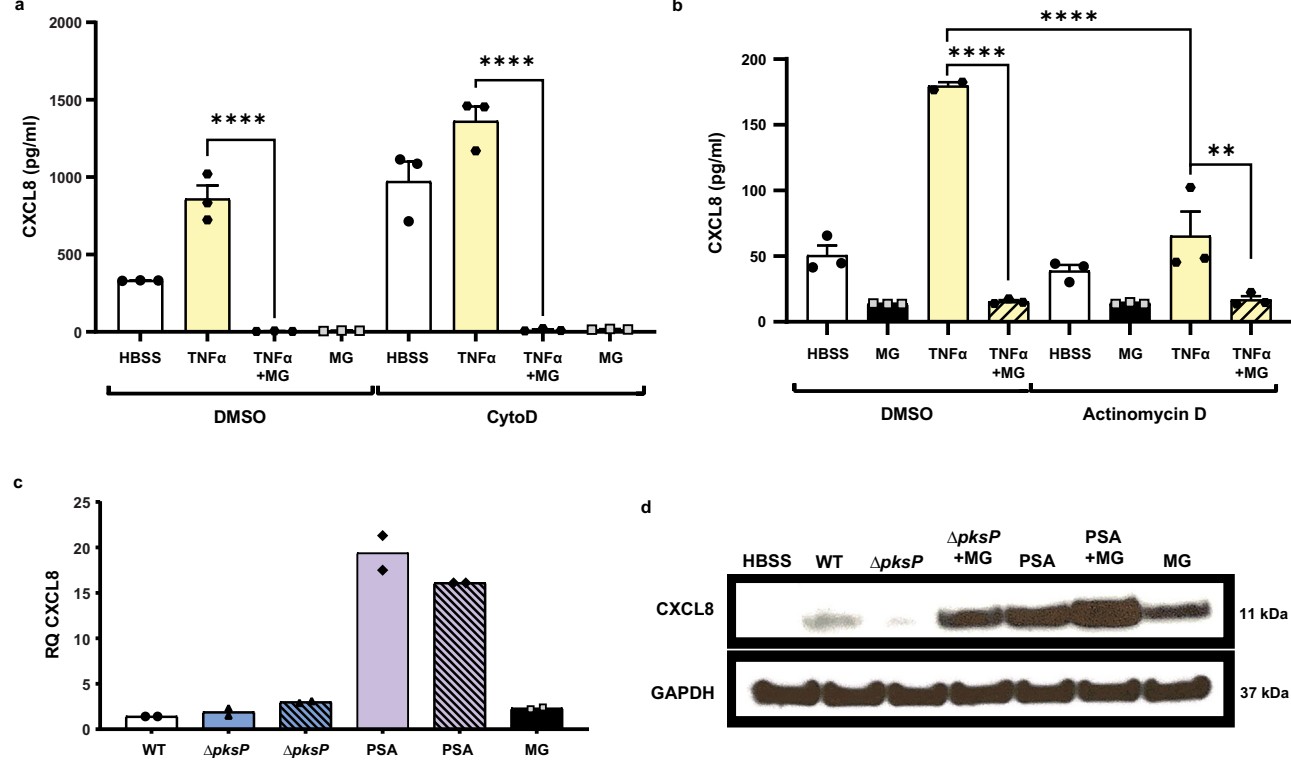

**Fig. 6 | Melanin suppression of chemokine secretion is independent of phagocytosis of melanin or de novo transcription, but exerts a post-translational blockade on CXCL8 secretion. a** H292 cells were incubated with DMSO vehicle control or cytochalasin D (CytoD; 20 µM) for 30 min, then stimulated with TNFα (100 ng/mL) and/or *Aspergillus* melanin ghosts (MG; $2 \times 10^7$/cm$^2$) for 6 h. Apical CXCL8 levels were measured by ELISA. $n = 3$ from three replicates; ****$p < 0.0001$. **b** H292 epithelium was exposed to Actinomycin D (1 µg/mL), then stimulated as described in A. CXCL8 apical secretion was measured using an ELISA assay. Two-Way ANOVA with Sidak's multiple comparisons test; Actinomycin D TNFα vs TNFα + MG ($p = 0.0112$); ****$p < 0.0001$. $n = 3$ from three replicates. **c–d** CXCL8 expression in H292 cells lysates was measured by (c) qPCR for RNA and (d) Western blot for intracellular protein, respectively, following 6 h stimulation with *A. fumigatus* wildtype (WT; B5233 strain; $1 \times 10^7$/cm$^2$), *ΔpksP* conidia ($1 \times 10^7$/cm$^2$) ± MG, *P. aeruginosa* PAO1 strain (PSA) ± MG, or MG alone ($5 \times 10^7$/cm$^2$). For Western blots, GAPDH was used as a loading control. See also Figure S5. $n = 2$ from technical duplicate. Data are represented as mean ± SEM. Source data are provided as a Source Data file.

sought to determine the biological role melanin played in innate immune responses. Here, we depict a critical role for fungal melanin in decreasing airway epithelial pro-inflammatory chemokine secretion via a post-translational mechanism, independent of both phagocytosis or de novo transcription. A recent study suggested depletion of extracellular CXCL10 and CCL20 chemokines by DHN melanin, but not IL-1α or CXCL8[19]. This study indicates that *Aspergillus* melanin binds to some pro-inflammatory cytokines in the extracellular space. While we demonstrate a reduction in CXCL8 and CXCL1 in our epithelial model, our data suggest that this is not largely in part to adsorption by melanin. The mechanisms responsible for recognition of fungal melanin by epithelial cells remains unknown and warrant further studies. MelLec (also known as Clec1A) is a CLR that recognizes *Aspergillus* DHN melanin, but not L-DOPA melanin of *C. neoformans*[31,69]. Both *A. fumigatus* DHN melanin and *C. neoformans* L-DOPA melanin block airway epithelial-derived inflammation suggesting a universal melanin effect at the epithelial surface. Given the specificity of MelLec for only DHN melanin, and the expression of human MelLec is restricted to endothelial cells and innate cells (Supplementary Fig. 1)[31], the respiratory epithelial response to melanin is likely independent of MelLec. Our experiments suggest that the phagocytosis of melanin particles is not necessary, since cytochalasin D does not block the ability of melanin ghosts to suppress epithelial CXCL8 secretion, suggesting that melanin is interacting with a cell surface receptor or a soluble factor released by epithelial cells. The observation that multiple forms of melanin retain the capacity to blunt the chemokine secretion by airway epithelial cells suggests that human pathogens exploit this pathway to

establish successful infection. Indeed, multiple organisms possess melanin including *A. fumigatus, C. neoformans*, and *Bordetella pertussis*[16,34,71]. Our understanding of melanin is evolving from a structural scaffold that provides protection from the harsh environment it inhabits to a multi-purpose molecule that disables the immune response at multiple steps. Our finding that *Aspergillus* melanin inhibits *P. aeruginosa*-induced airway epithelial chemokine secretion indicates that *Aspergillus* could play a role in dampening inflammation during co-infections common in CF patients. Colonization by *Aspergillus* could enhance the virulence of *P. aeruginosa* in patients by blocking early innate immune cell recruitment to sites of *P. aeruginosa* infection.

Our results indicate the suppression of early signaling cascades induced by fungal melanin, specifically calcium signaling. Calcium contributes through key signaling mechanisms in host immunity, such as TLR/CLR signaling, ROS production, and cytokine induction[72,73]. In innate immune cells, calcium has been linked to CXCL8 secretion in response to thapsigargin. Indeed, intracellular calcium stores in neutrophils are required and depletion of this calcium was not sufficient to trigger CXCL8 secretion[74]. Although this phenomenon has not been expanded to airway epithelial cells, thapsigargin increases CXCL8 in immortalized airway epithelium[75]. Prior studies have linked *Aspergillus* melanin to the sequestration of calcium inside the phagosome to block LC3-associated phagocytosis in monocytes[70], yet whether melanin impacted calcium-mediated secretion of cytokines and chemokines was not reported. Our data suggest that sequestration of calcium extends beyond inhibition of phagocytosis to include dampened

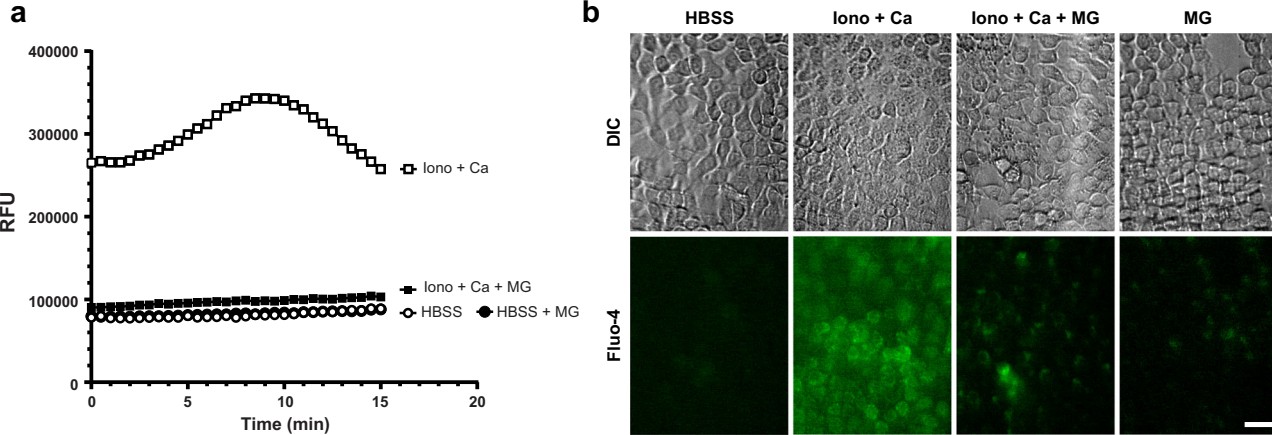

**Fig. 7 | Epithelial calcium fluxing dampened by fungal melanin. a** H292 cells were stimulated with media (HBSS) alone or melanin ghosts for 30 min followed by a 1 h incubation with ionomycin (2 μM) and calcium chloride (1.3 mM) for 2 h. Calcium fluxing measured by plate reader every 15 sec. Cell stimulated with media in the presence or absence of melanin ghosts produced virtually identical values. Data represent three technical replicates. **b** Representative DIC (top panel) and confocal (bottom panel) images of H292 calcium fluxing by fluo-4 in the presence or absence of melanin ghosts when treated with media alone (HBSS without calcium or magnesium) or ionomycin and calcium. Images from 10 min timepoint and representation of one region of interest in one set of experiments. Scale bar = 10 μm. Source data are provided as a Source Data file.

chemokine secretion. We propose that the melanin blunts calcium fluxing resulting in perturbation of actin polymerization and subsequent impairment in the release of CXCL1 and CXCL8. It is important to note that changes in actin polymerization may not be the sole contributor of calcium-mediated signaling in airway epithelial antifungal defense.

While we focused primarily on CXCL8 for this work, fungal melanin also inhibited the secretion of CXCL1 (Fig. 2b, Supplementary Figure 3a, and Supplementary Figure 4). While CXCL8 and CXCL1 are known neutrophil chemoattractants that play a role in epithelial-induced neutrophil transmigration other secreted compounds are also required for *P. aeruginosa* dependent neutrophil transmigration including hepoxilin A3[44–47]. Melanin ghosts dampened neutrophil transmigration in response to *P.aeruginosa*, indicating that the effect of fungal melanin may have a broad impact on airway epithelial cell secretion. Both CXCL8 and CXCL1 are secreted through the trans-Golgi network. The intracellular accumulation of CXCL8 protein with little detectable extracellular CXCL8 suggests that melanin may interfere with trans-Golgi secretion of CXCL8. Future studies will interrogate both traditional trans-Golgi secretomal pathways, as well as alternative secretion systems, such as IL-1β which is secreted via a gasdermin dependent mechanism, to determine the extent of the melanin secretion blockade.

Fungal melanins act as an anti-inflammatory agent at the respiratory epithelium, blocking post-translational secretion of chemokines to recruit host immune cells. *Aspergillus* melanin is required to establish invasive disease as the *ΔpksP* is avirulent in animal models of infection. The mechanism underpinning the virulence defect of the *ΔpksP* strain has long been attributed to melanin blockade of phagosomal maturation and ROS production by innate immune cells, critical requirements for host response to *Aspergillus*. However, our work highlights an additional role for melanin in the host-pathogen interaction, as an anti-inflammatory that prevents airway epithelial chemokine secretion thereby decreasing early recruitment of neutrophils to sites of infection. While there is a clear benefit to the pathogen in preventing inflammatory cell recruitment to the airways, there is a potential benefit to the host in dampening the host response to melanin at the airway epithelium. If melanin fails to silence inflammation in some patients, this could lead to chronic inflammatory conditions, including chronic pulmonary aspergillosis and allergic diseases such as allergic bronchopulmonary aspergillosis and fungal asthma.

## Methods

### Ethics Statement

All research presented complies with all relevant ethical regulations. Specifically, primary airway basal cells and neutrophils were isolated from healthy volunteer donors in accordance with Massachusetts General Hospital IRB-approved protocols (2002P002626 and 2015P000818, respectively). All individuals provided informed written consent prior to the collection of the sample. Consistent with IRB-protocol, sex/gender, age, and ancestry were not collected for these studies. All animal studies were conducted under protocols approved by the Institutional Animal Care and Use Committee Subcommittee on Research Animal Care at Massachusetts General Hospital (Protocol #2008N000078).

### H292 culture, upright and inverted Transwell culture

Human NCI-H292 pulmonary mucoepidermoid carcinoma cell line (H292) (American Type Cell Collection, #CRL-1848) was grown as previously described[28,47,66,67]. H292 cells were cultured in complete RPMI medium (cRMPI) (RPMI 1640 [Corning, #10-040-CV], L-glutamine [Gibco, #25030081], 10% heat-inactivated fetal bovine serum [FBS] (Gibco), and 1% penicillin/streptomycin [Gibco, #151410122]) at 37 °C in the presence of 5% CO$_2$. For upright Transwell culture, the apical compartments of Costar 6.5 mm or 12 mm Transwell® inserts with 0.4 μm pore polyester membrane (Corning, #38024 and #3460) were treated with a 60% Ethanol, 1:1000 rat tail collagen I (Corning, #354236) mixture and left open in tissue culture hood for at least 4 h to allow for evaporation. H292 cells were then liberated with 0.25% Trypsin-EDTA (Thermo Fisher Scientific, #25200072) and resuspended in cRPMI. The cell suspension was seeded onto the apical surface of the Transwells and cRPMI was added to the basolateral compartment. For inverted Transwell culture for use in neutrophil migration assays, Costar 6.5 mm Transwell® inserts with 3.0 μm pore polyester membranes (Corning, #3415) were inverted and treated with collagen. H292 cells in suspension were applied to the surface of the inverted Transwells to form a small tension bubble and incubated overnight. The following day, Transwells were placed upright in a Costar culture plate (Corning, #3473) and both the apical and basolateral compartments were filled with cRPMI. All transwell cultures were incubated at 37 °C in the presence of 5% CO$_2$ and used on day 7.

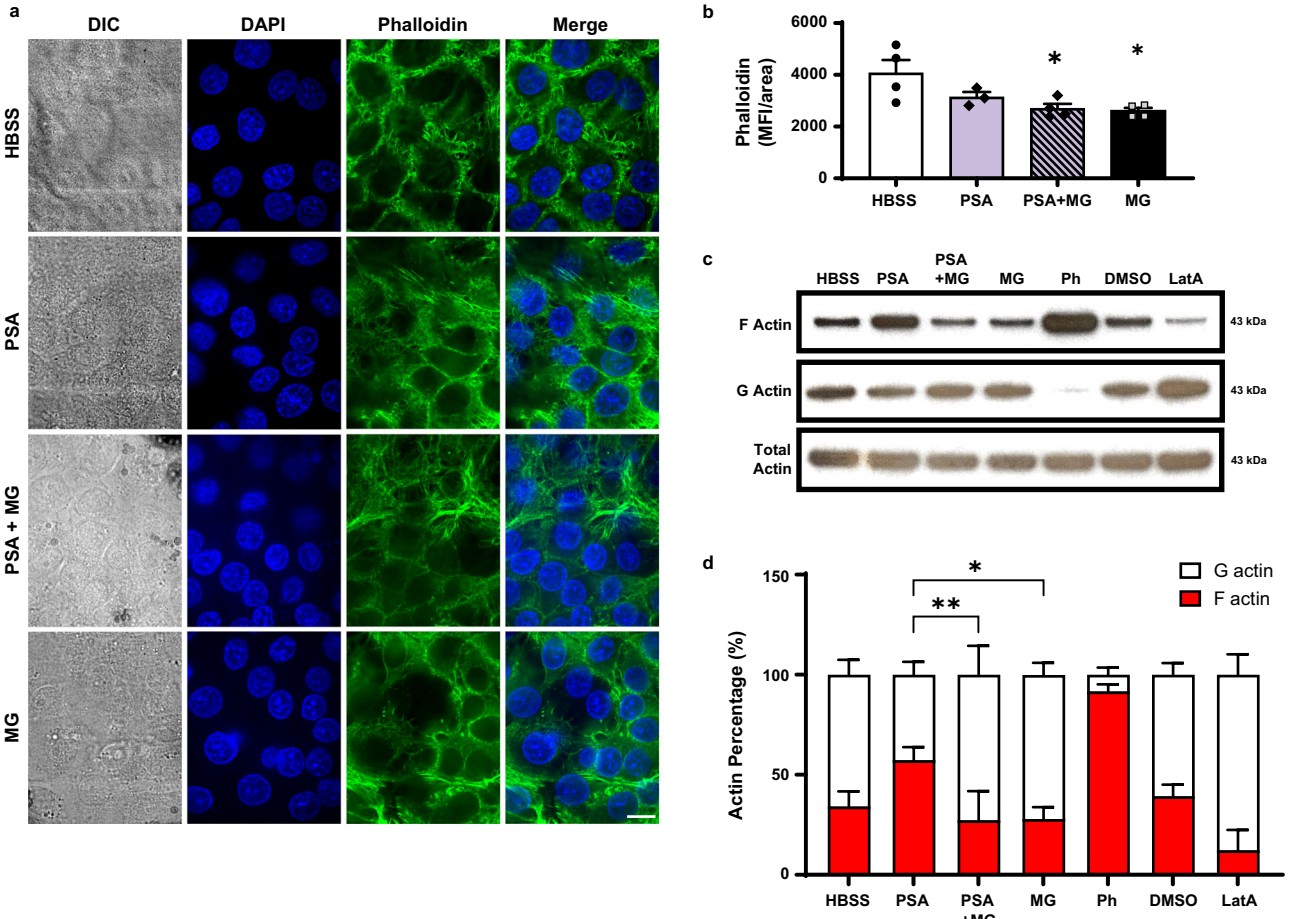

**Fig. 8 | Melanin disrupts actin structure in airway epithelial cells.**
**a** Representative images of phalloidin (green) and DAPI (blue) stained H292 cells stimulated with media alone (HBSS), *P. aeruginosa* PAO1 strain (PSA; $1 \times 10^7/$cm$^2$) ± MG, or MG alone ($5 \times 10^7/$cm$^2$) for 6 h. Images are a representation of one region of interest in one set of experiments Scale bar = 10 μm. **b** Mean fluorescence intensity (MFI) over area for phalloidin staining images (a). $n = 3$ regions of interest (PSA only) or $n = 4$ (HBSS, PSA + MG, MG). One-way ANOVA with Tukey's multiple comparisons test; *$p < 0.05$ compared to HBSS (PSA + MG [p = 0.0337]; MG

[p = 0.0231]). **c** Representative of F actin (top blot), G actin (1:20 dilution; middle blot), and total actin (bottom blot) measured by Western blot from three independent experiments. H292 cells were stimulated with medial alone (HBSS), *P. aeruginosa* PAO1 strain (PSA; $1 \times 10^7/$cm$^2$) ± MG, or MG alone ($5 \times 10^7/$cm$^2$), DMSO vehicle control, or latrunculin A (LatA) for 6 h. Phalloidin was additionally used as a positive control. (**d**) Quantification of three biological replicates F/G actin ratio from (b). N = 3; Two-way ANOVA with Tukey's multiple comparisons test; *$p = 0.0031$, **$p = 0.0025$. Source data are provided as a Source Data file.

## Primary human airway epithelial basal cell proliferation and upright and inverted ALI Transwell culture

Primary hAEC were grown as previously described[28,40,76–78]. Basal cells were cultured in small-airway epithelial cell medium (SAGM) (PromoCell, #C-21170) supplemented with 5 μM Y-27632 (Tocris, #1254), 1 μM A-83-01 (Tocris, #2939), 0.2 μM DMH-1(Selleck Chemicals, S7146), and 0.5 μM CHIR99021 (Tocris, #4423) and 1% penicillin/streptomycin on laminin-enriched 804G-conditioned media coated plates. For differentiation of basal cells into a fully pseudostratified epithelium at ALI, the apical compartments of Costar 6.5 mm or 12 mm Transwell® inserts with 0.4 μm pore polyester membrane (Corning, #38024 and #3460) were coated with 804G-conditioned media for at least 4 hours. Upon removal, a basal cell suspension in SAGM was applied to the apical compartment, and SAGM was added to the basolateral compartment and the transwells were incubated overnight. The following day, the SAGM was replaced with PneumaCult-ALI medium (StemCell, #05001) combined 1:1 with DMEM/F-12 (Gibco, #11320033) and incubated overnight. The media in the apical compartment was then removed to establish ALI. ALI hAEC were incubated for 16–21 days, wherein the apical compartment was kept dry, and the ALI media in the basolateral compartment was refreshed frequently.

For inverted ALI culture for use in neutrophil migration assays, Costar 6.5 mm Transwell® inserts with 3.0 μm pore polyester membranes (Corning, #3415) were inverted and treated with 804G-conditioned media. Basal cells in suspension in SAGM were applied to the surface of the inverted Transwells to form a small tension bubble and incubated overnight. The following day, Transwells were placed upright in a Costar culture plate (Corning, #3473) and both the apical and basolateral compartments were filled with SAGM. The following day, SAGM was replaced with PneumaCult-ALI medium combined 1:1 with DMEM/F-12 and incubated overnight. The media in the apical compartment was then removed to establish an air-liquid interface. The inverted ALI were incubated for 16–21 days wherein the apical compartment was kept dry and the media in the basolateral compartment was refreshed frequently. All ALI cultures were incubated at 37 °C in the presence of 5% CO$_2$.

## Fungal and bacterial culture

*Aspergillus fumigatus* strains *Af293*, *CEA10*, *B5233*, *ΔpksP*, and *ΔpksP* + pksP were used. The *B5233*, *ΔpksP*, and complemented *ΔpksP* + pksP strains were gifted by K.J. Kwon-Chung (National Institutes of Health; NIH). The *Af293* and *CEA10* strains were gifted by R.A. Cramer, Dartmouth College. All *A. fumigatus* strains were grown at 37 °C for

3–5 days on glucose minimal media slants (GMM)[79]. Conidia were harvested by applying a sterile solution of deionized water (Milli-Q, Millipore Sigma, Burlington, MA) containing 0.01% Tween 20 (Sigma-Aldrich, #P9416) to each slant followed by gentle surface agitation to liberate conidia using a sterile cotton-tipped swab. The solution was passed through a 40 μm cell strainer (CELLTREAT, #229481) to separate conidia from hyphal fragments. Conidia were then washed three times with sterile PBS and counted on a LUNA™ automated cell counter (Logos Biosystems). Conidia were used either on the day of harvest or stored at 4 °C overnight for use the following day. For in vitro experiments, conidia were applied to the apical surface. For animal experiments, conidia were used on the day of harvest.

*P. aeruginosa* strain PAO1 (ATCC, BAA-47) was grown overnight in 4 mL of LB liquid media (BD, #244520) at 37 °C with shaking. Cultures were inoculated from frozen stocks of PAO1 stored at −80 °C. After shaking overnight, 1 mL of overnight culture was centrifuged at 16,000 g x 5 minutes and washed three times in HBSS (StemCell, #37150). After the final wash, the *P. aeruginosa* was resuspended in 600 μl of HBSS. For in vitro stimulations, the *P. aeruginosa* was used at a 1:100 dilution. For quantification of bacterial number for in vivo experiments, the OD600 of a 1:10 dilution of bacteria in HBSS was obtained and bacteria were diluted appropriately. For in vitro experiments, *P. aeruginosa* was applied to the apical surface.

## Neutrophil isolation

Briefly, 50 mL of blood was drawn via venipuncture into a syringe containing acid citrate-dextrose. The blood was split between three 50 mL conical tubes and layered on top of a Ficoll-Paque solution (Cytiva Life Sciences, # 17144002), then centrifuged at 1500 g for 20 minutes at room temperature to obtain a buffy coat layer. Plasma and mononuclear cells were removed by aspiration. Red blood cells (RBC) were removed using 2% gelatin sedimentation technique followed by a wash with RBC lysis buffer. The remaining neutrophils were washed and resuspended in HBSS without calcium or magnesium (HBSS-; Thermo Fisher, #14175095) to a concentration of $5 \times 10^7$ neutrophils/mL.

## Neutrophil transmigration experiments

The neutrophil transepithelial migration experiments were performed as previously described (Fig. 1a)[28,40,41,67]. H292 cell monolayers or primary hAEC were grown on the underside of 3.0 μm pore size Transwell filters to enable neutrophil migration through the Transwell. Mature epithelium (at least 16 days at ALI for primary hAECs or day 7 monolayers of H292 cells) in 12-well format was used for all experiments. On the day of the experiment, Transwells were equilibrated in HBSS for at least 30 minutes prior to stimulation. Transwells were then inverted into a sterile 15 cm cell culture plate and infected on the apical surface. Cells were stimulated with media only (HBSS), *P. aeruginosa* (PAO1), *A. fumigatus* conidia, melanin ghosts, or melanin ghost plus 3000 pg/mL exogenous CXCL8 (R&D Systems, #208-IL). Infections were for 1 h on H292 cell monolayers and for 2 h on primary hAEC. After stimulation, the apical surface was washed and transwells were inverted into fresh 24-well culture plates containing HBSS alone or HBSS containing 100 nM N-formyl-methionyl-leucyl-phenylalanine (fMLP; uninfected positive control [Sigma Aldrich, #F3506]). Neutrophils were applied to the basolateral compartment and transmigration was allowed to proceed for 2 h (H292 cells) and 4 h (primary hAECs) at 37°C with 5% CO₂. After the migration period, the Transwells were discarded and the migrated neutrophils were quantified using a myeloperoxidase (MPO) assay. The migrated neutrophils in the apical compartment were lysed with 0.5% Triton X-100 (Sigma-Aldrich, #R8787-100ML) and neutrophil MPO activity was quantified using a colorimetric assay using citrate buffered 2,2′-azino-bis(3-ethylbenxothiazoline-6-sulfonic acid) diammonium salt (ABTS) solution (Sigma-Aldrich, #A9941-50TAB). After 10 min of incubation, the optical density at 405 nm was read using a spectrophotometer plate reader. In addition, to the experimental samples, a standard curve was prepared for each experiment was used to determine the number of migrated neutrophils based on the $OD_{405}$ value. Data are displayed as the number of transmigrated neutrophils, extrapolated from the standard curve.

## Melanin ghost preparation

Melanin ghosts were prepared using previously published methods[20,34–38]. In addition to melanin ghosts created in our lab using the following protocol, we also tested melanin ghosts created in the laboratory of Dr. George Chamilos with similar results to our own. Briefly, *A. fumigatus* strain B5233 was grown for 3 days in T75 flasks (CELLTREAT, #229341) on GMM at 30 °C. *C. neoformans* strain H99 (gift from Arturo Casadevall) was grown in liquid L-dopa media (15 mM glucose [Sigma-Aldrich, #G5767], 10 mM MgSO₄ [Sigma-Aldrich M2773-1KG], 29.4 mM KH₂PO₄ [Sigma-Aldrich, #P8281-100G], glycine 13 mM [Sigma, #G-7403], 3 μM vitamin B1 [Sigma-Aldrich, #T1270-25G], 1mM L-DOPA [Sigma-Aldrich, #333786]) and incubatemediad at 30 °C for 10–14 days. Flasks were seeded from frozen stocks maintained at −80 °C. To harvest fungal organisms, a sterile solution of deionized water (Milli-Q; Millipore Sigma) containing 0.01% Tween 20 (Sigma-Aldrich, #P9416-100ML) was added to each flask, and spores were liberated using gentle surface agitation with a sterile swab. The spore solution was passed through a 40 μm cell strainer to separate hyphal debris. To remove the fungal cell wall, spores were washed three times with sterile PBS, then resuspended in 10 mg/mL lysing enzymes from *Trichoderma harzianum* which has since been discontinued and replaced with 5 mg/mL lyticase from *Arthrobacter luteus* (Sigma-Aldrich, #L2524-50KU) dissolved in a 1 M Sorbitol (Sigma-Aldrich, S3889-500G) / 0.1 M Sodium Citrate (Sigma-Aldrich, S-4641) solution in deionized water. The solution was incubated in a 30 °C water bath for 24 h. To denature proteins, the spore solution was then washed twice with sterile PBS (Corning, #21-040-CM) and resuspended in a 4 M guanidine thiocyanate (Sigma-Aldrich, #50983-250 ML) solution and incubated at room temperature for 18 h with nutation. To hydrolyze proteins, spores were then washed twice with sterile PBS and resuspended in a 1.0 mg/mL Proteinase K (Roche Laboratories, #3115836001) in 10 mM Tris-HCl (Sigma-Aldrich, T3253-100G), 1 mM CaCl₂ (Sigma-Aldrich, #223506), and 0.5% Sodium Dodecyl Sulfate (Invitrogen, #15-525-017) pH 7.8, solution in distilled water. The suspension was incubated in a 65 °C water bath for 4 h, then washed three times with sterile PBS and resuspended in a 0.9% NaCl solution (Axell, YSPS0155). A Folch lipid extraction was then performed wherein the suspension was transferred to a separatory funnel along with methanol (Thermo Fisher Scientific, #02003345) and chloroform (Midland Scientific, #9180-22) and mixed vigorously. This was repeated twice, with the organic layer being drained each time and the soluble melanin-containing interfacial layer maintained. For the hydrolysis of residual acid-labile constituents, 6 M HCl (Sigma Aldrich, #T3253-100G) was added to the flask and boiled for 1 hour, then combined with equal volume sterile PBS to neutralize pH. The melanin ghost suspension was then washed with sterile PBS and placed in 0.22 μm centrifugal filter units (EMD Millipore, #UFC30GV0S). The melanin ghosts were then washed 3 times and resuspended in deionized water. The suspension was frozen overnight at −80 °C and lyophilized for two days. The melanin ghosts were finally resuspended in sterile PBS, sonicated, and counted on a LUNA automated cell counter (Logos Biosystems).

## Fungal-like particle (FLP) creation

We have previously created FLP using purified fungal carbohydrates, which we can associate with amine-coated polystyrene 3 μM microspheres (Polysciences, 17145-5) using either chemical conjugation or adsorption[57,58]. To create melanin-coated particles, synthetic melanin (Sigma Aldrich, M0418-100MG) was sonicated robustly in DMSO to create a fine suspension that was then incubated with amine-coated

polystyrene microspheres for 1 h at room temperature with shaking. The microspheres were washed copiously with PBS to remove all traces of DMSO (Sigma-Aldrich, #D2650) and free melanin particles. All of the microspheres obtained a brown coloration consistent with melanin association and there was no free melanin visible in the flow through from the FLP.

## ELISA

Mature epithelium (at least 16 days at ALI for primary hAECs or day 7 monolayers of H292 cells) in either 12 or 24-well format was used for all experiments. On the day of the experiment, growth media was removed and cells were equilibrated for at least 30 min in HBSS. HBSS was removed and replaced with fresh HBSS alone or containing the appropriate stimulants. Since the primary hAEC is pseudostratified epithelium, the concentration of applied stimuli was based upon the surface area of the epithelium rather than an MOI calculation. After application of stimuli, cell culture plates were spun gently at $250\,g \times 5$ min to bring pathogens in contact with epithelial surface. Stimulations were allowed to proceed for the desired time (4–24 h). Supernatants were harvested and centrifuged through Captiva 96 well 0.2 uM filter plate (Agilent Technologies, A5960002) at $625\,g$ for 10 min. Supernatants were frozen at −80 °C until use. Unless otherwise noted, supernatants measured were from the apical side. ELISAs were performed using either R&D Duoset kits (R&D Systems, #DY208 [CXCL8], #CY27505 [CXCL1]) or Biolegend LegendPlex assays (Legendplex™ Custom Panel (Human CXCL8, IL-8), #740399 and #741378) according to the manufacturer's instructions. For the R&D Duoset kits the ELISA was read using an i3X Spectrophotometer (Molecular Devices, LLC). For the LegendPlex assays, a BD FACSCelesta™ Flow Cytometer (BD) was used for data acquisition. The resultant flow cytometry files were then analyzed using the LEGENDplex™ Data Analysis Software Suite.

For cytochalasin D experiments, cytochalasin D (Sigma-Aldrich, #C8273-1MG) was added to HBSS media at a final concentration of 20 µM. Epithelial cells were incubated for 30 min at 37 °C prior to stimulation. At that time the media was exchanged for fresh HBSS containing the indicated stimulants in addition to 20 µM cytochalasin D. To inhibit transcription, actinomycin D (Sigma-Aldrich, #A1410-2MG) dissolved in DMSO was added to HBSS at a final concentration of 1µg/mL. An equal volume of DMSO in HBSS was added to the control wells. Epithelial cells were then incubated for 1 h at 37 °C prior to stimulation.

## In vivo neutrophil recruitment experiment

C57BL/6 mice of 6–12 weeks of age were obtained from Jackson Laboratories (#000664) and housed in the Massachusetts General Hospital-specific pathogen-free animal facilities (temperature ranges from 68–78 °F with humidity ranging from 30–70%; the facility has a dark/light cycle for 12 h [7AM–7PM=light, 7PM–7AM=dark]) for at least one week prior to infection. *Pseudomonas* strain PAO1 was diluted and quantified using OD600. Inoculums were prepared consisting of HBSS alone, $5.8 \times 10^5$ CFU PAO1 alone, $5.8 \times 10^6$ melanin ghosts alone, or a $5.8 \times 10^5$ CFU PAO1 combined with $5.8 \times 10^6$ melanin ghosts. Each inoculum was prepared in 50 µl total volume in HBSS. Mice were infected via oropharyngeal aspiration of the inoculum under isoflurane anesthesia. Infection was allowed to proceed for 4 h, at which time mice were euthanized using $CO_2$. Mice were exsanguinated, and the trachea was exposed and cannulated with a 22 G catheter (Exel Safelet Catheter, #14-841-20) that was secured in place using suture string. The lungs were flushed with a total of 2.3 ml of PBS containing 5% FBS. The collected bronchoalveolar lavage fluid was centrifuged at $300\,g \times 5$ min to pellet cells. The pelleted cells were resuspended in FACS buffer (PBS, 1% bovine serum albumin [BSA; Sigma Aldrich, #A7906-100G], 2 mM EDTA [Sigma Aldrich, #E7889], 0.1% sodium azide [Sigma Aldrich, #S2002-25G]) containing 1:100 dilution of Fc Block and a 1:250 dilution of Zombie Violet dye (Biolegend, #423113).

After incubation at 4 °C for 15 minutes, cells were centrifuged and then resuspended in FACS buffer containing an antibody cocktail consistent of anti-CD11b AF488 (Biolegend, #101217), anti-CD45 APC/Cy7 (Biolegend, #157618), anti-F4/80 AF647 (Biolegend, #123122), anti-SiglecF PE (Biolegend, #155506), and anti-Gr-1 BV650 (Biolegend, #108442). After incubation at 4 °C for 20 min cells were washed with FACS buffer and then fixed in 2% paraformaldehyde (Electron Microscopy Sciences, #15710) for 10 min at room temperature. Cells were washed and then resuspended in FACs buffer and stored at 4 °C (cells were stored no longer than overnight prior to flow cytometry). Splenocytes for a single mouse were prepared in parallel to use as controls for flow cytometry gating. Counting beads were added prior to flow and calibration beads were prepared using each of the fluorophores. Samples were run on a BD FACSCelesta™ Flow Cytometer, neutrophils were identified as $CD45^+F4/80^-SiglecF^-CD11b^+Gr-1^+$ cells, and analysis was performed using FlowJo 10 software (BD). Five to six mice were used per experimental group

## RNA extraction and qRT-PCR

The epithelium was grown on 0.4 µm pore-size Transwells until mature (at least 16 days hAEC or 7 days for H292 cells). On the day of stimulation, media was removed and cells were equilibrated in HBSS for at least 30 min prior to stimulation with either media alone (HBSS), *Pseudomonas aeruginosa* (PAO1), *A. fumigatus* conidia or melanin ghosts alone or in combination. Conidia were applied at $1 \times 10^7$ conidia per $cm^2$ and melanin ghosts were applied at 5X concentration at $5 \times 10^7$ conidia per $cm^2$. After 6 h of stimulation, supernatants were harvested for ELISA. At least 4 Transwells of epithelium were stimulated per condition, one well of epithelium was used for Western blot analysis and three wells were each harvested for RNA. For RNA prep, the epithelium was washed with ice-cold HBSS and then RNA was isolated using the RNeasy Mini Kit (Qiagen, #74134). After harvesting of cell lysates in QIAGEN RLT buffer containing beta-mercaptoethanol, cell lysates were transferred to QIAshredder (Qiagen, #79656) columns to homogenize the cells, the rest of the RNA extraction was performed according to RNeasy mini kit instructions. RNA was quantified using nanodrop. cDNA was created using the SuperScript IV VILO kit (Thermo Fisher Scientific, #11-756-050) according to the manufacturer's instructions. qRT-PCR was performed on cDNA using the Taqman® Fast Advanced Master Mix (Thermo Fisher Scientific, #4444557) and Taqman® Gene Expression assay probes CXCL8 labeled with FAM (Thermo Fisher Scientific, #4331182;Hs00174103_m1) and GAPDH labeled with VIC-MGB(Thermo Fisher Scientific, #448480;Hs99999905_m1). qRT-PCR reactions were run using the Applied Biosystems™ 7500 Fast Real-Time PCR System per kit instructions and analyzed using the ΔΔCt method.

## Western blot

Epithelium were prepared as described in "RNA extraction and qRT-PCR". After 6 h of stimulation, the epithelium was placed on ice and lysed with 1% NP40 lysis buffer containing protease inhibitor (Fluka Chemi, #74385). Following denaturation with NuPAGE LDS Sample Buffer (Thermo Fisher Scientific, #NP0007) proteins were resolved by SDS-PAGE (Thermo Fisher Scientific, #NP0002) on NuPAGE gels (Thermo Fisher Scientific, #NP0335BOX). Proteins were then transferred to a methanol-activated PDVF membrane (Perkin Elmer, #NEF1002001PK) using transfer buffer (0.025 M Tris, 0.192 M glycine, 20% methanol [Thermo Fisher Scientific, #LC3675]) and electrophoretic transfer at 100 V for 1 h. The membranes were blocked for 1 h at room temperature in 5% milk in PBS-0.01% Tween 20 (PBST), and then incubated with anti-CXCL8 antibody (Cell Signaling Technologies, #99407) overnight at 4 °C. The G-actin/F-actin assay biochem kit (Cytoskeleton, Inc., #BK037) was used for F/G actin fractionation experiments. Latrunculin A (Adipogen Life Sciences, #AG-CN2-0027-C100) at a concentration of 10 µM and 1:100 phalloidin were used as

positive controls for G actin and F actin, respectively. Phalloidin was added post lysate-collection as described in manufacture instructions for the kit. Membranes were washed and incubated for 1 h at room temperature with secondary HRP conjugated antibody (DAKO, #P0447). After washing, membranes were visualized using Western Lightning Plus ECL chemiluminescent substrate (Perkin Elmer, #NEL103001EA) and developed using BlueBlot HS Western Blotting Film (Fisher Scientific, #70474-01). Films were then scanned and processed using Adobe Photoshop 2021. Any contrast adjustments were applied evenly to the entire image and adhered to the standards set forth by the scientific community[80].

## Calcium flux plate assay

H292 cells were seeded into a 96-well plate at a density of $2 \times 10^4$ cells/well and incubated overnight at 37 °C with 5% $CO_2$ and 95% humidity. The cells were washed in serum-free cRPMI medium and treated with 5 nM Fluo-4 calcium indicator dye (Invitrogen, #F14201) with Pluronic F-127 (Invitrogen, #P3000MP) for 1 h at 37 °C with 5% $CO_2$ and 95% humidity. Subsequently, the cells were washed with HBSS without calcium and magnesium (Gibco, #14175-095) and pre-treated with *A. fumigatus* melanin ghosts or media (HBSS) alone for 30 min at room temperature. Melanin ghosts were removed by washing, and cells were stimulated with 2 μM ionomycin (Invitrogen, #I24222) and 1.3 mM calcium chloride (Fisher Bioreagents, #FLBP510500) for 2 h. Data was collected during the stimulation with 30 second intervals using a SpectraMax i3x Plate Reader (Molecular Devices, California, USA).

## Confocal imaging

For calcium flux assay, H292 cells were plated in Nunc Lab-Tek 8-chambered coverglass (Thermo Fisher Scientific, #155409) and allowed to adhere. Cells were grown in HBSS without calcium and magnesium. Fluo-4 in DMSO was added to cells for 1 h at 37 °C in the dark. Cells were washed and then stimulated with 2 μM ionomycin and 1.3 mM calcium chloride in the presence or absence of melanin ghosts. Cells were imaged every minute for 15 minutes (as described below).

For phalloidin staining, H292 cells were plated in Nunc Lab-Tek 8-chambered coverglass and allowed to adhere. Cells were stimulated for 6 h, washed with cold PBS, incubated with glyoxal solution for 1 h at room temperature, and quenched with 50 mM $NH_4Cl$ at room temperature. Cells were permeabilized with Triton X-100 for 4 min at room temperature. Non-specific binding in cells was prevented by incubating with a blocking buffer (5% gelatin from bovine skin [Sigma Aldrich, #G9391-100G]) for 15 min. Cells were incubated overnight at 4 °C with phalloidin iFlur 488 conjugate antibodies (Caymen Chemical, #20549) in dilution buffer (PBS, 1% BSA). Following three PBS washes, the chambered coverglass was then mounted with Vectashield Antifade Mounting Medium (Vector Laboratories, #H-1200) and Prolong antifade with DAPI (Molecular Probes, #P36962).

Images were captured on a Nikon Ti-Eclipse inverted microscope equipped with a CSU-X1 confocal spinning-disk head (Yokogawa, Sugarland, TX) and Coherent 4 W continuous-wave laser (Coherent, Santa Clara, CA) excited the sample. A high-magnification, high-numerical aperture objective (Nikon, 100x, 1.49 numerical aperture, oil immersion) was used. Images were obtained using an EM-CCD camera (Hamamatsu, Bridgewater, NJ). Image acquisition was performed using MetaMorph software version 7.10.5.476 (Molecular Devices, Downingtown, PA). Raw images were then cropped using Adobe Photoshop CS5 and assembled in Adobe Illustrator, version CS4 (Adobe Systems, San Jose, CA).

## Luminex assay

24-well Transwells were prepared using basal epithelial cells from three healthy human donors and differentiated for 16 days at ALI. On the day of stimulation, ALI media was removed and epithelium was equilibrated in HBSS media for 1.5 h prior to stimulation. The HBSS was removed and replaced with fresh HBSS media on the basolateral side, and 200 μL of HBSS media containing $1 \times 10^7$/cm² wild type conidia (strains *CEA10*, *Af293*, *B5233*), *ΔpksP* conidia, or *P. aeruginosa* strain PAO1. After 4 h, the apical and basolateral media was harvested and put through a Captiva 96 well 0.2 μm filter plate (Agilent Technologies, #A5960002) at 3000 rpm for 10 min. Supernatants were frozen at −80 °C. Data were processed using the Luminex analytical software. MILLIPLEX Map Human cytokine/chemokine Magnetic Bead Panel – Premixed 41 Plex (Millipore Sigma, #HCY-TOMAG-60K), run on a MAGPIX® xMAP instrument according to manufacturer's instructions.

## Pseudomonas epithelial association assay

H292 epithelium was infected with PAO1 alone or in combination with $5 \times 10^7$ *Aspergillus* melanin ghosts and incubated for 4 h. Wells were washed three times with HBSS to remove any nonadherent bacteria. To lyse epithelial cells, 1% Triton X-100 was added to each well, and plates were incubated with gentle agitation for 1 h at 4 °C. The lysate was collected and vortexed, and 1:10 serial dilutions were made in HBSS. 100 μL of each dilution was plated onto LB agar plates containing ampicillin and spread using glass beads. The plates were incubated at 37 °C overnight and colony forming units were counted the following day.

## LDH viability assay

The LDH viability assay was performed using CyQUANT LDH Cytotoxicity Assay (Invitrogen, #C20300) according to the manufacturer's instructions. After 6 h infection of H292 cells, a 1:10 dilution of 1% Triton X-100 was added to control well containing unstimulated epithelium to lyse the cells and incubated for 10 min at 37 °C 5% $CO_2$. 50 μL of supernatant from each experimental condition (all experimental conditions were performed in triplicate) was transferred to a 96 well plate and mixed with 50 μL of the kit Reaction mixture and allowed to incubate for 30 min at room temperature in the dark. Stop solution was then added to each sample and absorbance at 490 and 680 was measured using an i3X Spectrophotometer (Molecular Devices, LLC). LDH activity and % cytotoxicity were calculated according to assay kit instructions.

## Statistics and Reproducibility

All results were analyzed for statistical significance using GraphPad PRISM (GraphPad Software, 10.2.3). A p-value of 0.05 or less was considered significant. In studies with only two groups, an unpaired t-test was used. In studies with three or more groups, we used a one- or two-way ANOVA with multiple comparison tests (Tukey's, Sidak's, or Dunnett's). For in vitro experiments, no statistical method was used to predetermine sample size and we used a minimum of $n = 3$ for experiments based on prior work in the laboratory. For in vivo studies, a sample size of $n = 5$ mice per experimental condition was calculated based on prior studies conducted in our laboratory, the magnitude of the experimental effect (*e.g.*, the difference between test and control groups, which is also known as the expected effect size [Glass's delta]) ranges between 1.4 and >5.0 for the primary comparisons of the mouse experiments. An anticipated effect size of 2–5 can be detected with 79–99% power at the two-side 0.05 significance level. To ensure rigor, data was excluded for in vitro experiments if the transepithelial electrical resistance (TEER) was less than 1000 ohms on the day of the experiment. One data point was excluded in animal studies due to contamination in the sample. For reproducibility, all reported ELISAs and neutrophil transmigrations were done in both technical and biological triplicate. All qRT-PCR and Western blot experiments were performed in at least biological duplicates. Confocal imaging and calcium signaling were conducted in technical triplicates. Since all cells and animals were on the same background and received the same

treatments, the experiments were not randomized, and investigators were not blinded.

## Reporting summary

Further information on research design is available in the Nature Portfolio Reporting Summary linked to this article.

## Data availability

All data generated in this study are provided in the Supplementary Source Data file. Source data are provided in this paper.

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

## Acknowledgements

We would like to thank Drs. Bryan Hurley, Vincent Bruno, and Hongmei Mou for insightful discussions regarding our work. Additionally, we would like to thank Dr. George Chamilos for helpful discussions and providing melanin ghosts to test in parallel with those created in our own lab. This work was was supported by NIH/NIAID grant 1K08AI14755 (J.L.R), R01AI150181, R01AI136529, R21AI152499 (J.M.V),the MRC Center for Medical Mycology at the University of Exeter (MR/N006364/2 and MR/V033417/1), and the Wellcome Trust (102705, 217163) (G.D.B).

## Author contributions

Conceptualization: Jennifer Reedy, Kirstine Nolling Jensen, and Jatin Vyas. Investigation: Jennifer Reedy, Kirstine Nolling Jensen, Arianne Crossen, Kyle Basham, Chris Reardon, Hannah Brown Harding, Olivia Hepworth, Patricia Simaku, Geneva Kwaku, Kazuya Tone, Janet Willment, Delyth Reid, and Mark Stappers. Writing: Jennifer Reedy, Kirstine Nolling

Jensen, Rebecca Ward, Arianne Crossen, and Jatin Vyas. Manuscript review and revision: Arianne Crossen, Kyle Basham, Hannah Brown Harding, Chris Reardon, Olivia Hepworth, Patricia Simaku, Geneva Kwaku, Kazuya Tone, Janet Willment, Delyth Reid, Mark Stappers, and Gordon Brown. Resources: Jennifer Reedy, Jayaraj Rajagopal, Gordon Brown, and Jatin Vyas.

## Competing interests

The authors declare no competing interests.
