## [Peer Review File · Nature Communications]

Fungal melanin suppresses airway epithelial chemokine secretion through blockade of calcium fluxingREVIEWER COMMENTS

Reviewer #1 (Remarks to the Author):

In this well conducted study, the authors continue their previously published work (<https://doi.org/10.1128/IAI.00813-19>) to investigate the mechanism by which fungal melanin blocks the transmigration of neutrophils through airway epithelial cells. They elegantly show that melanin blocks CXCL8 secretion by epithelial cells. As this is a known neutrophil chemoattractant, it is presumed that this is the reason that impairs neutrophil recruitment.

Despite being a good study, in my opinion it does not add much knowledge to the previously published work. The authors had already established the methodology and, using it, detected that fungal melanin blocks neutrophil transmigration. Many of the results presented here add qualitatively to that previous conclusion (for instance, different types of melanin have the same effect, or the effect also affect transmigration in response to other challenges, as *Pseudomonas*), but do not really provide new information. The only novelty lies in the report that melanin blocks CXCL8 secretion, which even if interesting, does not constitute sufficient body of new work. The investigations should delve in much more detail in the underlying mechanism(s) and its consequences, as I suggest in some of my comments below.

- Although it is true that there is still much to learn about the functions of the airway epithelium in fungal infection, I think it is not fair to make statements as: "role of the respiratory epithelium in disease pathogenesis has not been defined" (lines 22-23) or that "there is a void in our understanding of the mechanistic interactions governing the invasion of *Aspergillus* at its first point of contact with the host, the airway epithelium (lines 48-49). These statements ignore or discredit many studies on the topic.

- If I understand correctly, the authors measured the production of many cytokines in response to Δ pksP conidia, but only show CXCL8 and CXCL1, as they state that they are the most abundant chemokines. The authors should show all the cytokines, as others might be relevant as well.

- Is it known if Δ pksP conidia trigger recruitment of more neutrophils to the alveolar space than wild-type conidia? Similar to the experiment presented in Fig. 3D, would addition of MG decrease the recruitment triggered by the infection with wild-type and mutant conidia?

- The comparative analyses throughout the manuscript should be done using ANOVA with multiple comparisons to consider the variability of the various samples. For instance, beta-glucan coated beads also decrease CXCL8 production in fig 4B, it is not correct to compare only PSA VS PSA+L-DOPA FLP without considering this action of PSA+BG FLP

- The different measurements of apical vs basolateral secretion is not clear. How is this experiment done? Maybe a schematic should be included as supplementary figure.

What type of secretion was measured in the previous experiments, apical or basolateral?

- It is not proven that reduction of CXCL8 is sufficient to prevent neutrophil transmigration. As MG blocks both CXCL8 production and transmigration, and this is a known chemoattractant it is an obvious hypothesis, but the potential role of other cytokines has been ignored. Does also CXCL1 play a role? Other cytokines? The authors should validate the results using a knock-out / silenced cell line.

- I do not understand the rationale to explain that melanin-induced transcription (is it described that melanin induces transcription??) would be relevant for the inhibition of CXCL8 production. I could understand the opposite, that melanin blocked transcription of this gene, but this is refuted in next section

- The finding that MG somehow impairs CXCL8 secretion from the cell is very interesting, potentially novel, but requires further validation and study.

How do the cells sense melanin? The authors discuss it is likely not via the MelLec receptor, but this should be further investigated. It is remarkable that mice deficient in MelLec (should not be able to detect melanin) recruited less neutrophils in BAL after *A. fumigatus* challenge (<https://doi.org/10.1038/nature25974>)

What is the transduction signalling cascade from the reception to the effect?

Is it specifically CXCL8 secretion impaired (unlikely), or is it a general action on cytoskeleton and/or vesicle transport? What processes are affected? Does this have other consequences for the cell?

Does this have any effect for the outcome of an infection? The melanin layer is present in conidia,

which are dormant structures, so it may be actually positive to dampen the infiltration of neutrophils when it is not required. In contrast, germinated conidia lose this layer and expose other PAMPs that strongly recruit neutrophils, does this bypass the blocking effect? All in all, this is a very interesting observation, but the mechanism(s) and relevance should be further investigated.

Reviewer #2 (Remarks to the Author):

In this manuscript, Reedy et al. describe a way in which fungal melanin interferes with a crucial part of the airway epithelium's response to fungal and bacterial pathogens. The paper is very well-written, very clear, and I found it enjoyable and smooth to read. I think the paper is complete and does not require any experiments, and the findings (and the discussion) will lead the way to future work by the authors to further unpack these interactions. Overall a great manuscript and I look forward to seeing the future research on the topic.

Title: Might be a little long and might bury the findings. Might benefit from rephrasing.

Line 21: Is *Cryptococcus* necessary in the abstract while most of the results concern *Aspergillus*?

Line 53-54: References?

Line 54-56: References?

Figure legends: I think cm² should be cm³ but might be better to use ml which is more standard, in my opinion

Line 210: tyrosinases and laccases – *C. neoformans* uses a laccase rather than a tyrosinase/phenol oxidase, and cannot convert tyrosine into melanin – only DOPA and catecholamines.

Line 211: physical structure or just talking about chemical structure?

Line 257: I am not sure if the publisher has a policy on data not shown, but if it is not allowed, then maybe you can restructure the sentence to cite other papers showing the limits/lack of melanin breakdown and degradation.

Line 262: Do you know if the CXCL8 production would return to normal if the melanin ghosts are removed?

Line 272: By any chance, did you confirm microscopically that no phagocytosis occurred?

Line 294/297: I think the reference to Figure 6D and 6C should be switched, if I am not mistaken.

Line 303: Is CXCL8 "production" interfered with? It seems to be produced okay, just the release is blocked.

Line 345: "necessary"

Line 351: Might include original references and/or additional fungal examples.

Figure 5A: The PSA and PSA+MG graph colors are a little difficult to differentiate. It is okay in the other figures because each bar is labelled.

Methods: Information on *Cryptococcus* culture strain/conditions?

Methods: If possible, use "g" instead of "rpm" for centrifugation speeds.

General: I think the manuscript is very clear and easy to understand, and the mechanism is explained well, but I think the manuscript would benefit from a simple summary figure, just to help the reader visualize the interactions better, or facilitate understanding.

General: I think Nature requires figures to include individual data points in the graphs, so turning the bar graphs into showing just the data points and error bars or including dots overlaid on the bar graph might be necessary.

General: I like how there is mention of pros/cons/considerations of one of the culture model used.

Reviewer #3 (Remarks to the Author):

In this manuscript from Reedy et al, it is shown that melanin, both DHN and L-DOPA forms, are able to "deplete" CXCL1 and CXCL8 from the cell supernatants and prevent PMN migration in both in vitro and in vivo infection systems. The data presented by the authors is robust and strong, demonstrating that the effects are not occurring through melanin uptake or transcriptional changes, but rather likely due to acting a sponge for the chemokines. That said the major weakness of the manuscript is that the authors do not show a cytokine/chemokine that is not altered by the presence of the melanin ghost -- are there cytokines unaffected by melanin? Moreover, this finding is interesting in the context of the recent work (published after this would have submitted) from Vincent Bruno and colleagues that saw melanin depleting CXCL10 and CCL20, but not IL-1alpha and CXCL8 in their system -- similarities and differences of these two systems/papers should be discussed, both taken together both papers support this novel role for melanin in being immunomodulatory. Overall, this is a well constructed studies with some minor weaknesses that can be addressed to highlight a novel role for melanin in modulating host immunity.

Specific Points:

- 1) Major point - Are there any cytokines/chemokine not depleted by the melanin ghost?
- 2) Figure 4B - Why were DHN melanin FLPs not constructed and used in these studies as well?
- 3) Figure 4 - I believe an ANOVA test would be the appropriate statistical test for this figure rather than a t-test since there are multiple groups that should be being compared to a common control group.
- 4) Figure 2 - Shouldn't the pksP-null strain be compared to the B5223 parental strain since that is the background of the mutant, as stated in the text?
- 5) Minor point - Figure 2 - Data presentation style should be changed. Have 2 panels across and 3 panel down so that each chemokine is a column and each experiment is a row.
- 6) Figure 6A - Why does CXCL8 expression go up in the HBSS group treated with CytoD? Also, is the data similar for all of Figure 6 for CXCL1?

Thank you for giving us the time to address the concerns raised by the reviewers. We also thank the reviewers and editor for their valuable comments and suggestions. After completing several new experiments, we have modified the manuscript (NCOMMS-23-14269) accordingly and incorporated the changes in the revised text. The revised manuscript was reviewed for grammatical and spelling mistakes. Given the new data provided, we have added 9 new authors: Kirstine Nolling Jensen (now co-first author), Olivia W Hepworth, Patricia Simaku, Geneva Kwaku, Kazuya Tone, Janet Willment, Delyth Reid, Mark Stappers, and Gordon Brown. All changes in the manuscript are notated in blue font.

Reviewer #1

1) *Despite being a good study, in my opinion it does not add much knowledge to the previously published work. The authors had already established the methodology and, using it, detected that fungal melanin blocks neutrophil transmigration. Many of the results presented here add qualitatively to that previous conclusion (for instance, different types of melanin have the same effect, or the effect also affect transmigration in response to other challenges, as Pseudomonas), but do not really provide new information. The only novelty lies in the report that melanin blocks CXCL8 secretion, which even if interesting, does not constitute sufficient body of new work. The investigations should delve in much more detail in the underlying mechanism(s) and its consequences, as I suggest in some of my comments below.*

Response: Our new data significantly expands upon our initial submission to demonstrate a clear mechanism of how melanin blocks secretion. Specifically, the presence of fungal melanin blocks calcium fluxing in airway epithelium, as shown by a calcium flux assay and live cell imaging in our new Figure 7. Inhibition of calcium fluxing results can impact numerous functions of cells, including actin polymerization. Thus, we investigated actin structure by phalloidin staining using confocal microscopy and biochemical analysis examining F/G actin ratios. Our new data unveiled restructuring of the actin cytoskeleton and direct impact on actin polymerization (new figure 8). These new data are incorporated in the abstract (lines 37-38), results (lines 333-356), discussion (lines 418-432), and methods (lines 813-816; 825-860).

2) *Although it is true that there is still much to learn about the functions of the airway epithelium in fungal infection, I think it is not fair to make statements as: “role of the respiratory epithelium in disease pathogenesis has not been defined” (lines 22-23) or that “there is a void in our understanding of the mechanistic interactions governing the invasion of Aspergillus at its first point of contact with the host, the airway epithelium (lines 48-49). These statements ignore or discredit many studies on the topic.*

Response: We have now modified the text to clarify the gaps in the field and ensure that selected prior publications within the field are highlighted.

3) *If I understand correctly, the authors measured the production of many cytokines in response to Δ pksP conidia, but only show CXCL8 and CXCL1, as they state that they are the most abundant chemokines. The authors should show all the cytokines, as others might be relevant as well.*

Response: We conducted a multiplex assay for multiple cytokines and determined that melanin-induced changes occurred for only CXCL8 and CXCL1. Supplementary Figure 2 shows these new data.

- 4) *Is it known if Δ pksP conidia trigger recruitment of more neutrophils to the alveolar space than wild-type conidia? Similar to the experiment presented in Fig. 3D, would addition of MG decrease the recruitment triggered by the infection with wild-type and mutant conidia?*

Response: Prior work by our new authors (PMID: 29489751) demonstrated that the *pksP* conidia recruited more neutrophils after 4 h of intratracheal inoculation compared to wildtype *A. fumigatus* (comparison of extended data 8d vs. 8j in this paper). We selected *Pseudomonas* since the bacteria more potently induced CXCL8 in our *in vitro* system when compared to *Aspergillus*. We increased our n for our *in vivo* infection experiment with *Pseudomonas aeruginosa* to measure the recruitment of immune cells (Fig 3d). These data represent a general phenomenon not specific to fungi.

- 5) *The comparative analyses throughout the manuscript should be done using ANOVA with multiple comparisons to consider the variability of the various samples. For instance, beta-glucan coated beads also decrease CXCL8 production in fig 4B, it is not correct to compare only PSA VS PSA+L-DOPA FLP without considering this action of PSA+BG FLP.*

Response: We have reviewed the statistical analyses for all figures and updated the statistical tests accordingly. With respect to Figure 4b, we did statistical analysis by one-way ANOVA for BG FLP but found no statistical significance.

- 6) *The different measurements of apical vs basolateral secretion is not clear. How is this experiment done? Maybe a schematic should be included as supplementary figure.*

Response: We now include a schematic in Figure 1 to demonstrate our strategy for an inverted air-liquid interface, which denotes the apical and basolateral spaces used for all experiments. Additionally, the text (lines 136-137) was amended to reflect this change and better describe our culture method.

- 7) *What type of secretion was measured in the previous experiments, apical or basolateral?*

Response: Unless otherwise stated (e.g., apical vs basolateral experiments), all studies measured cytokine secretion from the apical compartment. We updated the results, methods, and figure legends to ensure clarity of the type of secretion measured.

- 8) *It is not proven that reduction of CXCL8 is sufficient to prevent neutrophil transmigration. As MG blocks both CXCL8 production and transmigration, and this is a known chemoattractant it is an obvious hypothesis, but the potential role of other cytokines has been ignored. Does also CXCL1 play a role? Other cytokines? The authors should validate the results using a knock-out / silenced cell line.*

Response: We thank the reviewer for their helpful comment. To account for changes to the secretion of other cytokines, we employed a multiplex assay of pro-inflammatory cytokines. Only CXCL1 and CXCL8 were different when comparing melanized and non-melanized forms of *A. fumigatus* (new Extended Data Figure 2). Additionally, rather than using a knockout cell line, we conducted a migration assay in the presence of MG ghosts with or without exogenous CXCL8 to determine if the defect in migration was due to a lack of secretion or absorption. Our data revealed that exogenous CXCL8 increased neutrophil recruitment across the epithelium, suggesting that CXCL8 is not sequestering secreted CXCL8 to prevent recruitment of neutrophils. A new figure panel is included in the main text (Fig. 2g; lines 171-176).

9) *I do not understand the rationale to explain that melanin-induced transcription (is it described that melanin induces transcription??) would be relevant for the inhibition of CXCL8 production. I could understand the opposite, that melanin blocked transcription of this gene, but this is refuted in next section.*

Response: We apologize for the confusion. We have reworded the text to accurately reflect that the presence of melanin does not block the transcription of CXCL8 but rather the suppression of chemokine secretion occurs through the inhibition of calcium fluxing (lines 306-309).

10) *The finding that MG somehow impairs CXCL8 secretion from the cell is very interesting, potentially novel, but requires further validation and study. How do the cells sense melanin? The authors discuss it is likely not via the MelLec receptor, but this should be further investigated. It is remarkable that mice deficient in MelLec (should not be able to detect melanin) recruited less neutrophils in BAL after *A. fumigatus* challenge (<https://doi.org/10.1038/nature25974>).*

Response: We now include data to demonstrate that the MelLec receptor is not expressed by airway epithelial cells by Western blot. We utilized an antibody developed by Gordon Brown, Kazuya Tone, Janet Willment, Delyth Reid, and Mark Stappers. These individuals were included on the authors' list for their assistance with these data. These new data are included in Extended Data Figure 1 and the results (lines 124-127). Our results presented in this manuscript also suggest that melanin is exerting early effects in epithelial cells and thus could occur prior to detection by cells that express MelLec. Furthermore, while we unveil a role in epithelial responses, the effects seen in MelLec deficient rodents could occur due to expression on other cells in the lung (*i.e.*, endothelial and immune cells). The identification of a putative cell surface receptor for fungal melanins on lung epithelial cells is beyond the scope of the current study.

11) *What is the transduction signalling cascade from the reception to the effect? Is it specifically CXCL8 secretion impaired (unlikely), or is it a general action on cytoskeleton and/or vesicle transport? What processes are affected? Does this have other consequences for the cell?*

Response: To further examine signaling cascades modulated by fungal melanin, we conducted a series of experiments to dissect the potential mechanism responsible for the dampened secretion. First, we looked at calcium signaling as an early event that modulates numerous antifungal responses. New data unveiled *Aspergillus fumigatus* melanin ghosts mute calcium fluxing. Although calcium contributes to many pathways, we focused on actin structure as a potential mechanism to impaired CXCL8 secretion. We examined actin polymerization by phalloidin staining and F/G actin ratio through confocal microscopy and biochemical analysis, respectively. As predicted by the reviewer, melanin ghosts disrupt the actin structure. These data on calcium and actin structure are presented in Figure 7 and Figure 8, respectively.

12) *Does this have any effect for the outcome of an infection? The melanin layer is present in conidia, which are dormant structures, so it may be actually positive to dampen the infiltration of neutrophils when it is not required. In contrast, germinated conidia lose this layer and expose other PAMPs that strongly recruit neutrophils, does this bypass the blocking effect?*

Response: The $\Delta pksP$ mutant has been demonstrated as a less virulent strain, as demonstrated by improved survival (PMID: [9393803](https://pubmed.ncbi.nlm.nih.gov/31111111/)). Based on our and other laboratories' prior work, we hypothesize that the exposure of PAMPs in $\Delta pksP$ mutants enables a swift immune response

to clear the pathogen, while melanized conidia can evade detection of PAMPs for longer, thus establishing an infection. The work in this manuscript focuses on the molecular mechanisms underpinning fungal melanin; a dedicated study on the impact of fungal melanins on infection outcomes is the next logical step.

Reviewer #2:

- 1) **Title: Might be a little long and might bury the findings. Might benefit from rephrasing.**
Response: We have revised the title to “Fungal Melanin Suppresses Airway Epithelial Chemokine Secretion Through Blockade of Calcium Fluxing”.
- 2) **Line 21: Is *Cryptococcus* necessary in the abstract while most of the results concern *Aspergillus*?**
Response: We updated the abstract accordingly and ensured that no reference to *Cryptococcus* is included.
- 3) **Line 53-54: References? Line 54-56: References?**
Response: We have now added references to both lines.
- 4) **Figure legends: I think cm² should be cm³ but might be better to use ml which is more standard, in my opinion**
Response: For the airway epithelial model, the importance lies in the area (not volume) that the cells are in contact with the inoculum on the apical surface. Thus, the surface area (cm²) remains the best measure for the figure legends rather than volume. Indeed, this is the model used by our lab (PMID: 31767773).
- 5) **Line 210: tyrosinases and laccases – *C. neoformans* uses a laccase rather than a tyrosinase/phenol oxidase, and cannot convert tyrosine into melanin – only DOPA and catecholamines.**
Response: We amended the text to clarify that L-DOPA melanin occurs through two pathways, tyrosinase and laccase, depending on the starting molecule. Furthermore, we highlighted that *Cryptococcus* utilizes only laccase to synthesize melanin.
- 6) **Line 257: I am not sure if the publisher has a policy on data not shown, but if it is not allowed, then maybe you can restructure the sentence to cite other papers showing the limits/lack of melanin breakdown and degradation.**
Response: We have reworded the text accordingly (lines 281-282) and added new references (reference numbers: 19, 62, 63).
- 7) **Line 262: Do you know if the CXCL8 production would return to normal if the melanin ghosts are removed?**
Response: Although we would like to do this experiment, the melanin ghosts are too sticky to be confident that all particulate have been removed.
- 8) **Line 272: By any chance, did you confirm microscopically that no phagocytosis occurred?**
Response: Yes, we confirmed by confocal microscopy and observed no phagocytosis at the 6-hour timepoint.

9) *Line 294/297: I think the reference to Figure 6D and 6C should be switched, if I am not mistaken.*

Response: We agree and have now amended the text to refer to Figure panels 6c and 6d appropriately.

10) *Line 303: Is CXCL8 “production” interfered with? It seems to be produced okay, just the release is blocked.*

Response: We revised the text accordingly to clarify that CXCL8 secretion, rather than production, is inhibited.

11) *Line 345: “necessary”*

Response: Thank you for identifying this typo. We have revised the text accordingly.

12) *Line 351: Might include original references and/or additional fungal examples.*

Response: We agree and now include the primary articles for the listed pathogens (reference numbers 16, 34, 71).

13) *Figure 5A: The PSA and PSA+MG graph colors are a little difficult to differentiate. It is okay in the other figures because each bar is labelled.*

Response: We updated the color scheme to better differentiate the graph colors, specifically, we selected a lighter shade to ensure that the black lines are easily distinguished for MG-treated groups. Since we utilized the same colors across all figures, we have updated all figures to match this new color scheme.

14) *Methods: Information on Cryptococcus culture strain/conditions?*

Response: We updated the Methods section to include information on the *C. neoformans* strain (H99) and growth conditions under the heading “Melanin ghost preparation”.

15) *Methods: If possible, use “g” instead of “rpm” for centrifugation speeds.*

Response: We have revised the text accordingly to describe centrifugation speeds as g rather than rpm.

16) *General: I think the manuscript is very clear and easy to understand, and the mechanism is explained well, but I think the manuscript would benefit from a simple summary figure, just to help the reader visualize the interactions better, or facilitate understanding.*

Response: We agree with the reviewer and can include a new schematic if requested by the editor.

17) *General: I think Nature requires figures to include individual data points in the graphs, so turning the bar graphs into showing just the data points and error bars or including dots overlaid on the bar graph might be necessary.*

Response: We reviewed the guidelines provided by *Nature* and now include individual data points on all bar graphs of the main and supplementary data figures.

18) *General: I like how there is mention of pros/cons/considerations of one of the culture model used.*

Response: We thank the reviewer for their positive comment.

Reviewer #3:

1) Major point - Are there any cytokines/chemokine not depleted by the melanin ghost?

Response: We conducted a multiplex assay to assess multiple chemokines and cytokines. Please refer to the response to Reviewer #1, comment 3 for details of the experiment and new data.

2) Figure 4B - Why were DHN melanin FLPs not constructed and used in these studies as well?

Response: Unlike L-DOPA melanin, which is monomeric, DHN melanin is a polymer. Thus, we are unable to construct the FLPs as done in this and previous studies.

3) Figure 4 - I believe an ANOVA test would be the appropriate statistical test for this figure rather than a t-test since there are multiple groups that should be being compared to a common control group.

Response: We have reviewed the statistical analyses for all figures and updated the statistical tests accordingly.

4) Figure 2 - Shouldn't the pksP-null strain be compared to the B5223 parental strain since that is the background of the mutant, as stated in the text?

Response: Correct. We use the parental B5223 strain in all comparisons with the $\Delta pksP$ strain of *A. fumigatus*. In addition to these strains, we examined the impact of MGs from multiple *A. fumigatus* strains (i.e., CEA10 and Af293) to ensure effects were not strain-specific in Figure 2. Otherwise, all wildtype *A. fumigatus* refer to the B5223 strain. We reviewed the text, figures, and figure legends and revised them to ensure clarity of the strain used in each figure.

5) Minor point - Figure 2 - Data presentation style should be changed. Have 2 panels across and 3 panel down so that each chemokine is a column and each experiment is a row.

Response: We have updated the figure accordingly.

6) Figure 6A - Why does CXCL8 expression go up in the HBSS group treated with CytoD? Also, is the data similar for all of Figure 6 for CXCL1?

Response: These data confirm prior studies that found treatment of CytoD increased basal CXCL8 secretion in H292 cells (PMID: [16499908](https://pubmed.ncbi.nlm.nih.gov/16499908/); Figure 2B). We amended the text to reference this paper. We did not investigate CytoD treatments, actinomycin D treatments, RNA levels, or intracellular protein levels of CXCL1.

REVIEWERS' COMMENTS

Reviewer #1 (Remarks to the Author):

I would like to commend the authors for the great effort they have done to respond to my comments. All my doubts are clarified, and the article now presents truly new mechanistic data. It is a great study.

I only have a couple of minor points related with the new data, mostly to improve clarity of the presented results.

1. In figure 7a, there are 4 groups in the legend, but only 3 are visible in the graph.

2. Figure 8a. The difference in fluorescence is quite subtle here. Can the authors add a quantification of the signal to demonstrate a significant effect?

3. Figure 8b&c. The authors show a representative blot, and I guess a representative quantification of the ratio actin F/G. Can a quantification of the three independent experiments be done and shown? This would better reflect the magnitude and the variability of the effect (even allow statistical analysis)

Reviewer #3 (Remarks to the Author):

The authors addressed all my original points. The newly added data on the role of melanin in disrupting Ca^{2+} signaling and actin filamentation are a nice new addition to the manuscript to extend the study beyond previously published work.

We thank the reviewers and editor for their valuable comments and suggestions. We have modified the manuscript (NCOMMS-23-14269) accordingly and incorporated the changes in the revised text. All changes resulting from reviewer comments in the manuscript are notated in blue font.

Reviewer #1

1) *In figure 7a, there are 4 groups in the legend, but only 3 are visible in the graph.*

Response: We acknowledge that our HBSS and HBSS+MG are overlapping, and thus, look like there are only three groups in Figure 7a. To improve visibility, we increased the size of the black circles for the HBSS+MG group and moved the legend text to the end of each corresponding line. This overlap is also addressed in the figure legend.

2) *Figure 8a. The difference in fluorescence is quite subtle here. Can the authors add a quantification of the signal to demonstrate a significant effect?*

Response: We have now added a graph demonstrating the quantification of phalloidin fluorescence (new 8b) to reflect this observation.

3) *Figure 8b&c. The authors show a representative blot, and I guess a representative quantification of the ratio actin F/G. Can a quantification of the three independent experiments be done and shown? This would better reflect the magnitude and the variability of the effect (even allow statistical analysis)*

Response: We have updated the quantification graph to include three biological replicates and now include stats. Please note that now these figures are 8c and 8d.